# Household Rituals and Merchant Caravanners: The Phenomenon of Early Bronze Age Donkey Burials from Tell eṣ-Ṣâfi/Gath, Israel

**DOI:** 10.3390/ani12151931

**Published:** 2022-07-28

**Authors:** Haskel J. Greenfield, Jon Ross, Tina L. Greenfield, Shira Albaz, Sarah J. Richardson, Aren M. Maeir

**Affiliations:** 1Department of Anthropology, St. Paul’s College, University of Manitoba, Winnipeg, MB R3T 2M6, Canada; tlgreenfield@gmail.com (T.L.G.); srichardson162@gmail.com (S.J.R.); 2Department of Land of Israel Studies and Archaeology, Ariel University, Ariel 40700, Israel; jon.ross521@gmail.com; 3Martin (Szusz) Department of Land of Israel Studies and Archaeology, Institute of Archaeology, Bar-Ilan University, Ramat Gan 5290002, Israel; shirakisos@gmail.com (S.A.); arenmaeir@gmail.com (A.M.M.)

**Keywords:** ritual, sacrifice, Early Bronze Age, southern Levant, Near East, zooarchaeology, *Equus asinus*, donkey, animal figurines, building foundation deposits, trade, merchant homes

## Abstract

**Simple Summary:**

The goal of this study is to increase our understanding of the role of ritual in the domestic residences of commoners in early complex societies in the ancient Near East. Most archaeologists have concentrated their research on rituals taking place in the public and administrative areas of early cities (e.g., temples and palaces). However, the bulk of the population lived in simple domestic residences and were not involved in the public ritual displays except as onlookers. We present the results of our recent excavations at the Early Bronze Age site of Tell eṣ-Ṣâfi/Gath, Israel. Our excavations at the site have uncovered the remains of early domestic donkeys and other goods that were buried as the neighbourhood was constructed and houses were renovated. These provide insight into the role of ritual in everyday life for most people in these early cities. As the donkey burials are very limited in their location in these early cities, we propose that such residences were inhabited by merchant families.

**Abstract:**

Most studies of ritual and symbolism in early complex societies of the Near East have focused on elite and/or public behavioural domains. However, the vast bulk of the population would not have been able to fully participate in such public displays. This paper explores the zooarchaeological and associated archaeological evidence for household rituals in lower-stratum residences in the Early Bronze Age (EB) of the southern Levant. Data from the EB III (c. 2850–2550 BCE) deposits excavated at the site of Tell eṣ-Ṣâfi/Gath, Israel, are illustrative of the difficulty in identifying the nature of household rituals. An integrated analytical approach to the architecture, figurines, foundation deposits, and domestic donkey burials found in lower-stratum domestic residences provides insights into the nature of household rituals. This integrated contextual perspective allows the sacred and symbolic role(s) of each to be understood and their importance for EB urban society to be evaluated.

## 1. Introduction

The goal of this paper is to investigate ritual and symbolic behaviours that occur within non-elite (lower stratum) households in early urban societies in the Near East. Data from our recent excavations of an Early Bronze Age (EB) (Note: The abbreviations of EBA and EB are used in different ways in this paper, largely according to their conventional usage in the literature of the southern Levant. EB is used when we refer to a phase or series of phases within the Early Bronze Age (e.g., EB II–III or EB III), while EBA refers to the period as a whole) III domestic residential neighbourhood at Tell eṣ-Ṣâfi/Gath, Israel are utilised to illustrate non-elite ‘ritual life’ and performance, both of which are poorly understood within the EB urban populations. This holistic perspective includes a discussion of not only the traditional realms of archaeology (e.g., architecture, artefacts) but also the zooarchaeology of the site. The results are subsequently grounded within a larger discussion on the nature of household rituals in the southern Levant during the EB.

The discussion centres on the excavation and analytical results of five complete and several incomplete burials of domestic asses found under the floors of private residential buildings exposed in an EB merchant neighbourhood (Area E). The unusually large number of donkey burials from a very limited number of EB buildings is unprecedented and poses interesting questions regarding the identity of the inhabitants and the ritual, symbolic, and economic role(s) of asses in early urban society. Why were so many donkeys buried in an urban environment/context in this small part of the settlement at Tell eṣ-Ṣâfi/Gath? Was this practice simply a form of ‘disposal’ for beloved pets and work animals buried without ceremony, or did it have other meanings? While the ritual character of some donkey burials from EB excavations has lately been called into question [1,2], only a holistic treatment of the asinine burials from Tell eṣ-Ṣâfi/Gath may allow for a fuller investigation of the different ritual roles (sacred, symbolic, and economic) of asses in EB III households. Therefore, this paper provides an opportunity to clarify the character of the donkey burials recovered from Area E, with regards to the intersection of ritual life, symbols, and the economy (outside of a temple and funerary setting) in a residential EB neighbourhood. Such opportunities to investigate household rituals are rare during the EB of the Levant since visible nonelite groups and practices are traditionally overshadowed by the focus on the priestly class in towering temples.

## 2. Cult, Symbols, and Ritual in Archaeology

Symbolic behaviour (e.g., cult and ritual practices and their associated symbols) are inherent to daily economic and social life and play a fundamental role in structuring and organising society, e.g., [3,4,5,6,7,8,9,10]. However, the query of such symbolic behaviour is far from a simple and straightforward field of study. Exactly what qualifies as cult and ritual (how it is defined and the interpretation of meaning) and how it should be studied (interpretive frameworks) is the subject of continuous impassioned debate, particularly with regards to the linkages between ritual and society (mechanisms for societal change) [3,7,11,12,13,14].

The archaeological study of cult and ritual is often neglected (considered a low priority), due in large part to “definitional uncertainty” and ambiguity over how best to conceptualise and study (*objectify and materialise*) cult and ritual objects. This is compounded when limited to (or constrained by) the surviving material residues that make-up/(re)constitute the archaeological record [3,7,15]. This situation has clearly changed in recent years, as more and more archaeologists have taken up the challenge [16,17,18]. Regardless of definitions, ritual behaviour is highly complex, for it encompasses the intangible, the ‘transcendent’, and the ‘indefinable’ [7,19]. Christopher Hawkes, for instance, considered ritual life and ‘spiritual beliefs’ as the most difficult and problematic subject to access in archaeology and placed it at the top of his ‘ladder of inference’ [20]. This strong pessimism/scepticism was pervasive and encouraged by one of the foremost Assyriologists of the 20th century, A. Leo Oppenheim. He stressed the fragmentary and indirect nature of the available evidence deemed inadequate for forming a picture of vanished polytheistic religious practices that are far removed from the present experience of monotheistic religion in the modern world [21,22]. The conceptual chasm was too vast to span. Cult and ritual were placed beyond the conventional reach/ambit of the discipline and remained exiled to the margins/periphery.

Archaeologists are trapped by a powerful paradox and (the all-too-familiar) materialist dilemma because the ritual landscape is far from a wholly physical one. Yet its study in archaeological contexts is entirely constrained by the surviving material residues. How we materialise the immaterial at this point becomes an epistemological (and ontological) Gordian Knot. The iceberg analogy [23] to understanding the archaeological record is probably most apt—very little is preserved and/or visible of behaviour, but this is what we need to reconstruct ultimately. Taphonomy, both cultural and natural, as well as the immaterial nature of most behaviour, prevents the easy reconstruction of much of the archaeological record.

For these reasons, the very terms ‘ritual’ and ‘ceremonial’ were (and still are) frequently invoked by archaeologists in ways that were/are problematic (and simplistic) to reference the weird and wonderful or the odd and unexplained (fetishizing the exotic). Yet many dimensions of material culture can be subsumed under cult and ritual and the material implications are ‘profound’ [3,5,8,9,13,24,25,26]. Attitudes and paradigms in archaeological thought have swung from ignoring and downplaying cult, ritual, and ‘symbol-based approaches’ (such as in both processual and culture-historical frameworks where it was a low priority and considered ‘epi-phenomenal’ and/or a ‘mentalist’ preoccupation/distraction) to treating religion and ritual as singular and near totalising (well exemplified by traditional scholarship on Mesopotamian temple estates and the social organisation of early urban societies in the 4th and 3rd millennium BCE of the Near East [22,27,28]). A new blend of determinism (and causality) has surfaced in recent scholarship emphasising the centrality of cult and religion for driving unprecedented change (*Neolithisation*) following the Pleistocene. Hence, the new provocative rallying cry: ‘it all began with ritual’ [6,19,25,29,30], even though there are claims for complex cultures prior to the Holocene [31].

Nowhere else in the world have these themes/issues/debates been so intensely scrutinised than Hodder’s 25-year Neolithic excavation project at Çatalhöyük [5,32,33], followed by Schmidt’s excavations at Gobekli Tepe [24,25,26]. In a recent volume dedicated to the memory of Klaus Schmidt, Hodder [19] defines religion as “a “transcendental social”, an imagined communal identity of a social entity” realised/expressed through ritual theatre. The impact of religion on society is signalled by highly repetitious and habitual social practice not conditioned by material/functional requirements (enduring continuity in the layout and use of space over time), combined with meaningful delineation of place (boundary markers) and numerous acts of commemoration and remembrance. Such commemorative acts typically include the frequent occurrence of foundation deposits and related activities attached to “the ending and starting of buildings”, as well as feasts, the curation of ‘heirlooms’, the deliberate ‘deposition of things’, the continuity of iconographic traditions with highly charged symbolism, and the like [5]. These indicators are not dissimilar from the generalised categories of traits listed by traditional cross-cultural approaches to the archaeology of cult and religion, e.g., [13,18].

Hodder takes his treatment of religion and ritual a step further by introducing the concept of ‘history making’ [3,5]. History-making is composed of concerted acts to consolidate and intensify a community’s historical ties and ‘attachment to place’ through ritual (achieving a heightened sense of temporal depth and ‘long-term memory construction’), to reinforce and expand social ties and networks in the present. Thus, fostering greater cooperation and solidarity for economies that were becoming increasingly dependent on delayed returns for labour investment. Hodder distinguishes two forms of ‘history-making practices’ held in constant tension and negotiation: (1) practices limited to house-based descent groups (as is well exemplified by Çatalhöyük, and which are more dominant in the PPNB after sedentarisation was complete even though it is doubtful if “sedentarisation’ was ever “complete”—rather it became a dominant component of society); and (2) practices carried out at the level of the collective by solidarity-based groups that converged on ‘public’ buildings as is well exemplified by Gobekli Tepe, and which are more dominant in the PPNA in the earlier stages of sedentarisation). Both groups/entities invested in religious and ritual practices that involved ‘history making’. These dynamics open up a fresh perspective for making sense of ritual practice and the general relationships between material culture and religion that extend far beyond the Neolithic world of southwest Asia [30].

## 3. The EB of the Southern Levant

The EB of the southern Levant is divided into four major periods—EB I–IV. With the advent of EB I (c. 3600–3100 BCE—the abbreviations CE and BCE are substituted in this essay in place of the more commonly used AD and BC to follow the convention used in the southern Levant), and synchronous with the Late Uruk period of Mesopotamia), we see the rise of the first urban fortified centres that come to dominate the region. These are in fact secondary states, as the primary states of Mesopotamia and Egypt probably influence the development of southern Levantine urbanism (Figure 1). Over time, such centres come to dominate almost every part of the coastal plain and hill country of the southern Levant during EB II (c. 3100–2900 BCE) and EB III (2900–2500 BCE). In the subsequent EB IV/Intermediate Bronze (IB) (2500–2000/1900 BCE), the regional system of fortified urban centres collapses. It is not until the subsequent Middle Bronze (MB) period (c. 2000–1750 BCE) that urban lifestyles are renewed in the region.

While there is a long running debate concerning whether the large fortified EB sites were in fact truly urban and state-level [34,35,36], recent regional settlement patterns and other evidence from archaeological excavations (monumentality, town planning, and craft economies/specialisation) have largely put this issue to rest [37,38,39]. It is now widely accepted that the city-state is based on a centralised authority within a multitiered social and political hierarchy, with a top-down redistributive economy [40,41]. Thus, during EB II–III, there is increased (vis-à-vis earlier periods) intensification of agriculture [42,43], trade and exchange across the region [44,45,46], productive specialisation [47,48,49,50,51], and economies characterised by delayed returns for high-labour investments [37,52,53,54].

## 4. The Ritual Landscape of the EB Southern Levant

Discussion of EB cult and ritual in early urban Levantine societies has mostly followed cultural–historical and processual lines of thought, centred on typing similarities and differences in building plans, with respect to the evolution/development of sacred monumental architecture for time–space systematics [39,55,56,57,58,59,60,61,62]. Attention has primarily focused on:Early sacred public architecture—iterations of elite monumentality and collective expressions of institutionalised ritual and solidarities that crosscut individual households [63];Mountain cult sites and open-air sanctuaries in remote locations presumably servicing pastoral nomads [64]; andTombs and mortuary/funerary practices (e.g., Bab edh-Dhra, Jericho, and the *nawamis* in the Sinai) [65,66];

EB temples and cultic compounds, such as the Area J temple sequence at Megiddo or the temple sequence at Ai, are highly visible in the archaeological record and leave a prominent material footprint/trace. They constitute the most conspicuous manifestation of ritual behaviour in the archaeological record of the Early Bronze. However, they do not provide a full picture of the sacred landscape and lack a full in situ cultic assemblage. There are no surviving examples of cult statues found in situ (i.e., standing in niches or altars within the temples or other buildings) in EB southern Levantine sites.

The temples themselves are found abandoned and empty of most of the valuable cult objects/contents. Rarely are cult statues found, particularly outside of Mesopotamia. In the southern Levant, representations and depictions of (anthropomorphic) deities, such as the EB II cult stelae from Arad, are few and far between and may not be deities at all [67]. There is a long history of stelae in the region extending back into the PrePottery Neolithic and down into the EB [68], but none have inscriptions. As a result, the identity of the major EB deities and their specific religious rites/ceremonies is conjectural and not securely known [63,67]. There is continued debate over whether smaller cultic buildings/shrines operating in early urban contexts (e.g., the so-called ‘Twin Temples’ at Arad or the White Building at Yarmuth) functioned as communal/public shrines or were simply ‘domestic chapels’ and/or ‘patrician houses’ in EB II–III [58,59].

Consequently, cult and ritual life are notoriously difficult to access and investigate in the absence of written records, artistic representations/depictions, and overt cultic equipment/realia (shrines, altars, podiums, *masseboth*, inscribed stele etc.) [69]. Hence, research on the EB of the southern Levant in general continues to generally favour/privilege the ritual performance of emergent elites (the top end of the social hierarchy) in early urban contexts because of their visibility (e.g., monumental architecture)—they have more and unique paraphernalia that is therefore more archaeologically visible. Even more apparent is that ritual practices (and by extension ‘history-making’) are poorly understood (and virtually non-existent) for house-based descent groups (most of the population) at early urban centres. There are no texts and limited iconography to supplement the meagre archaeological record. Unlike the Neolithic, this demographic/group/substratum of early urban society is ordinarily very difficult to target and access with regards to cult and ritual practices, particularly in the Early Bronze Age Levant. The dataset of equid burials from Tell eṣ-Ṣâfi/Gath provides a rare instance wherein it is possible to access the ritual life and practice of nonelite social groups and households (at a major early urban community) operating outside of traditional elite temple contexts and highly centralised institutions.

## 5. Material—The EB at Tell eṣ-Ṣâfi/Gath

The site of Tell eṣ-Ṣâfi/Gath is located on the western border of the Judean foothills and overlooks the main east–west pass through the Elah valley connecting the central hills with the southern coastal plain (Figure 1 and Figure 2). A substantial fortification system rings the site (Figure 3). At 24 hectares in size, Tell eṣ-Ṣâfi/Gath is the same size, if not larger than, other fortified EB III urban centres in the regional settlement system, such as at Erani (c. 25 hectares) and Yarmuth (c. 18 ha.) [52,53,54].

Excavation on the eastern slope (Area E) focused on the exposure of a late EB III neighbourhood (Figure 4), with significant exposure of remains belonging to Strata E5a, E5b, and E5c (Figure 5, Figure 6, Figure 7, Figure 8, Figure 9 and Figure 10) [70,71,72,73]. The rich pottery assemblage from the E5 strata is chronologically diagnostic of the EB IIIC repertoire, and radiocarbon dating for the termination of the final EB phase indicates a date range of ca. 2550–2600 cal. BC [70,71,73].

## 6. Materials and Results—Evidence for Ritual in the EB at Tell eṣ-Ṣâfi/Gath

### 6.1. Architecture as Evidence for Ritual

Three rectilinear rows of buildings were exposed in Area E at Tell eṣ-Ṣâfi/Gath (Figure 4). They are separated by a narrow street and gridded out on a NW to SE orientation. They are clearly part of a larger neighbourhood that encompasses much of the east end of the site, as the same orientation for building layout is found in excavations almost 100 m distant on the site [74].

**Figure 4 animals-12-01931-f004:**
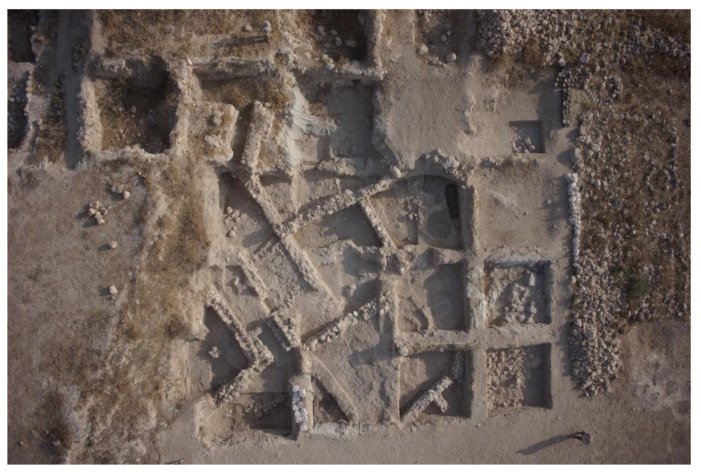
Aerial photograph of Area E excavations (2015) showing an outline of some of the EB buildings. Site north is at top of photograph and in the following illustrations. Copyright @ Tell eṣ-Ṣâfi/Gath Archaeological Project.

The Stratum E5 buildings undergo three phases of construction and renovation characterised as three strata: reconstruction of the neighbourhood (Stratum E5c/earliest), followed by two phases of renewal (Stratum E5b/middle and Stratum E5a/latest) before the site is abandoned c. 2550 BCE. Buildings are given a new building number in subsequent phases of occupation. For example, Building 134307 in E5c becomes Building 74512 in E5b and Building 74505 in E5a. This allows us to distinguish deposits associated with each phase of occupation—i.e., construction and renovation. (Figure 5, Figure 6 and Figure 7). The three E5 Strata include walls and floors from three phases, each one reusing earlier wall stubs, with minor modification to building plans (the addition of partition walls, installations, and floor renewals), thus preserving the overall layout of the neighbourhood over time. The exterior walls of buildings were built directly on top of earlier walls in the same alignment, with narrow rooms flanking the alleyway and connecting to open courtyards.

**Figure 5 animals-12-01931-f005:**
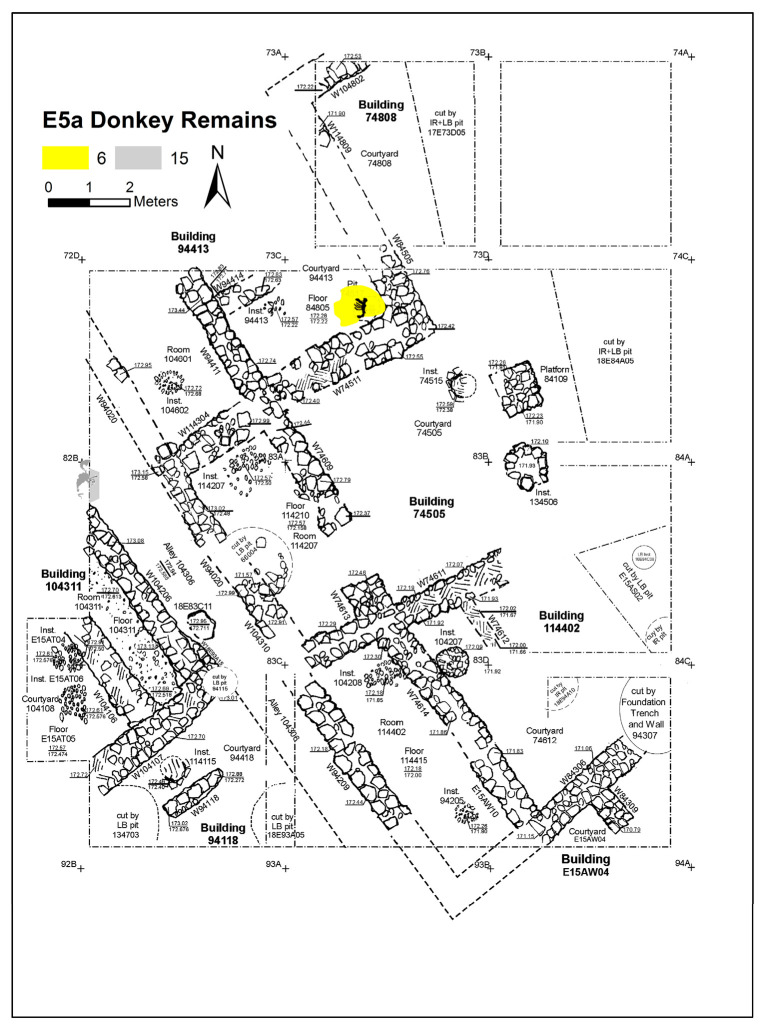
Plan of the EB III neighbourhood uncovered in Stratum E5a of Area E, Tell eṣ-Ṣâfi/Gath, showing the location of the donkey burials. Copyright @ Tell eṣ-Ṣâfi/Gath Archaeological Project.

**Figure 6 animals-12-01931-f006:**
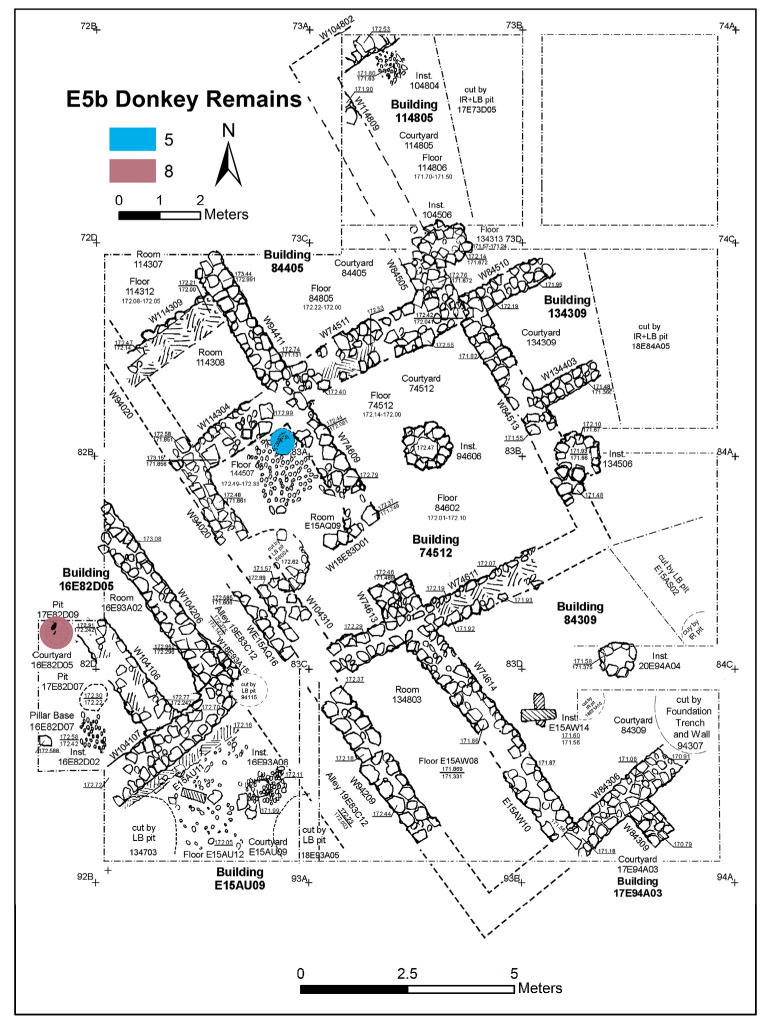
Plan of the EB III neighbourhood uncovered in Stratum E5b of Area E, Tell eṣ-Ṣâfi/Gath, showing the location of the donkey burials (L144511 and L17E82D09). Copyright @ Tell eṣ-Ṣâfi/Gath Archaeological Project.

**Figure 7 animals-12-01931-f007:**
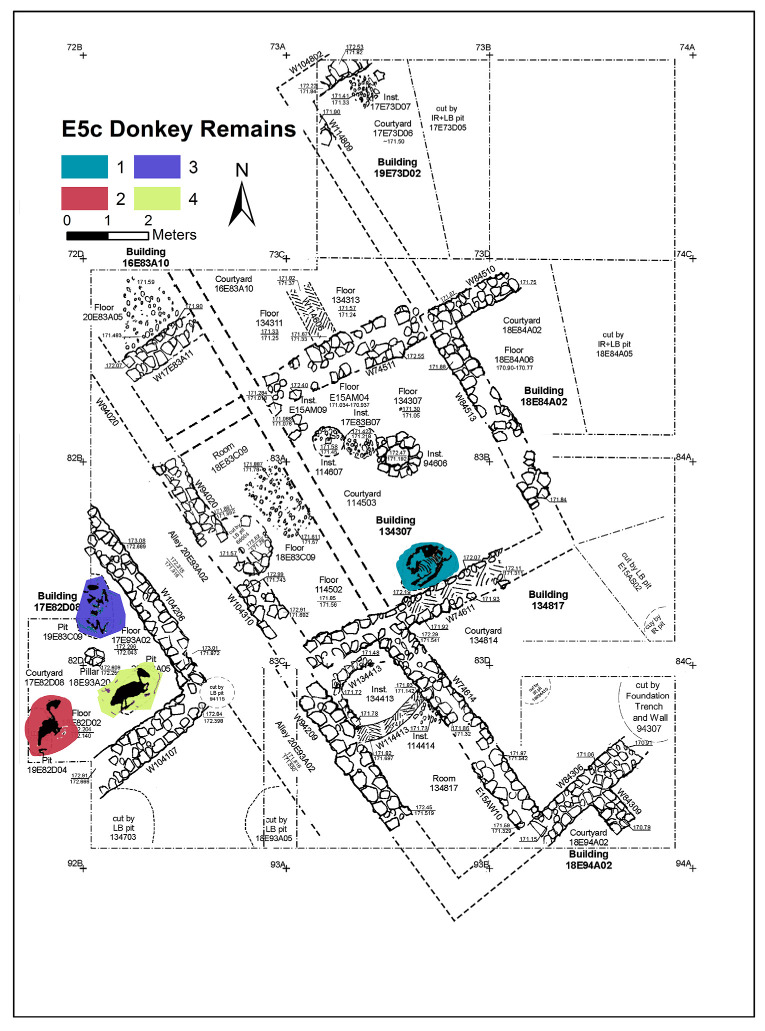
Plan of the EB III neighbourhood uncovered in Stratum E5c of Area E, Tell eṣ-Ṣâfi/Gath, showing the location of the donkey burials (L114506, L19D82D04, L19D83C09, and L20D93A05). Copyright @ Tell eṣ-Ṣâfi/Gath Archaeological Project.

The Stratum E5a-c buildings are small but sturdy multi-roomed units consisting of a courtyard and an ancillary room on a rectangular plan and sharing parti-walls with the neighbouring building. The buildings are remarkably uniform in their layout and continue to be so in each phase.

Comparable southern Levantine EB house plans include Stratum IIIc1 (EB IIIA) in Area F and IIIb2 (EB II) in Trench III at Jericho [75,76], Level G2 at Yarmuth [77], Period C in Area EY at Tel Beth Yerah [78,79], Stratum 19 in Field X at Tall al-‘Umayri (Jordan) [80], and Phase II in Area 2 at Tell Abu al-Kharaz [81,82].

All the above, in addition to the cultural continuity displayed by the ceramic inventories through each phase, are strong evidence for residential and cultural continuity over time in the neighbourhood. While the exposures for the underlying (Strata E6–E9—Figure 8, Figure 9 and Figure 10) were too limited to determine the full nature of continuity in the architectural footprint from the very beginning of the neighbourhood (Stratum E9), there appears to be no significant change in the overall orientation, layout, and material culture character of the neighbourhood over time. Overall, the EB III occupation in Area E at the site was long and dense, as indicated by the depth of a probe in Square 93B (c. 2.5 m below Stratum E7) excavated in the final season of 2017—it showed a long succession of surfaces and floor renewals from the E5 to the E9 (Figure 10). At the bottom of the probe, ceramics from the EB II were recovered [83].

Generally, the buildings appear to be non-elite/lower-stratum domestic residences. All material remains within the Stratum E5a-c and underlying strata are related to the household consumption of food or daily refuse and contain a variety of everyday items, including both mundane and exotic trade goods [70,71,72,73,84].

**Figure 8 animals-12-01931-f008:**
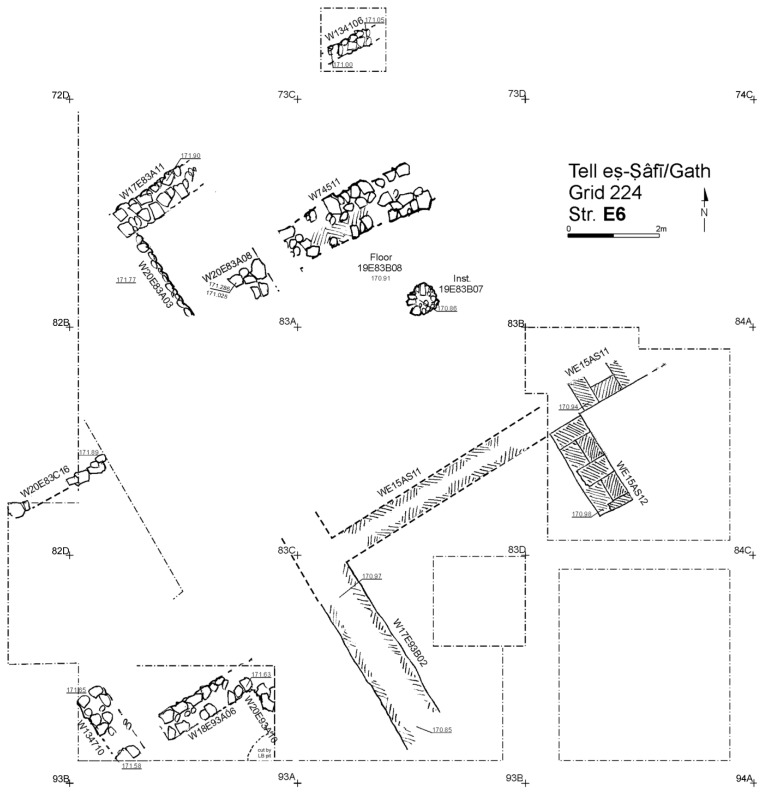
Plan of the EB III neighbourhood uncovered in Stratum E6 of Area E, Tell eṣ-Ṣâfi/Gath. Copyright @ Tell eṣ-Ṣâfi/Gath Archaeological Project.

**Figure 9 animals-12-01931-f009:**
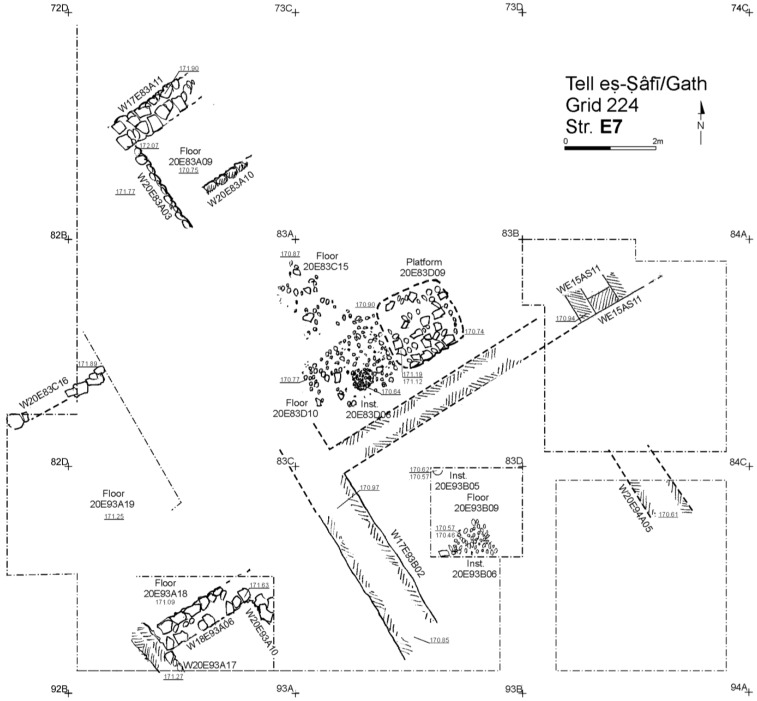
Plan of the EB III neighbourhood uncovered in Stratum E7 of Area E, Tell eṣ-Ṣâfi/Gath. Copyright @ Tell eṣ-Ṣâfi/Gath Archaeological Project.

**Figure 10 animals-12-01931-f010:**
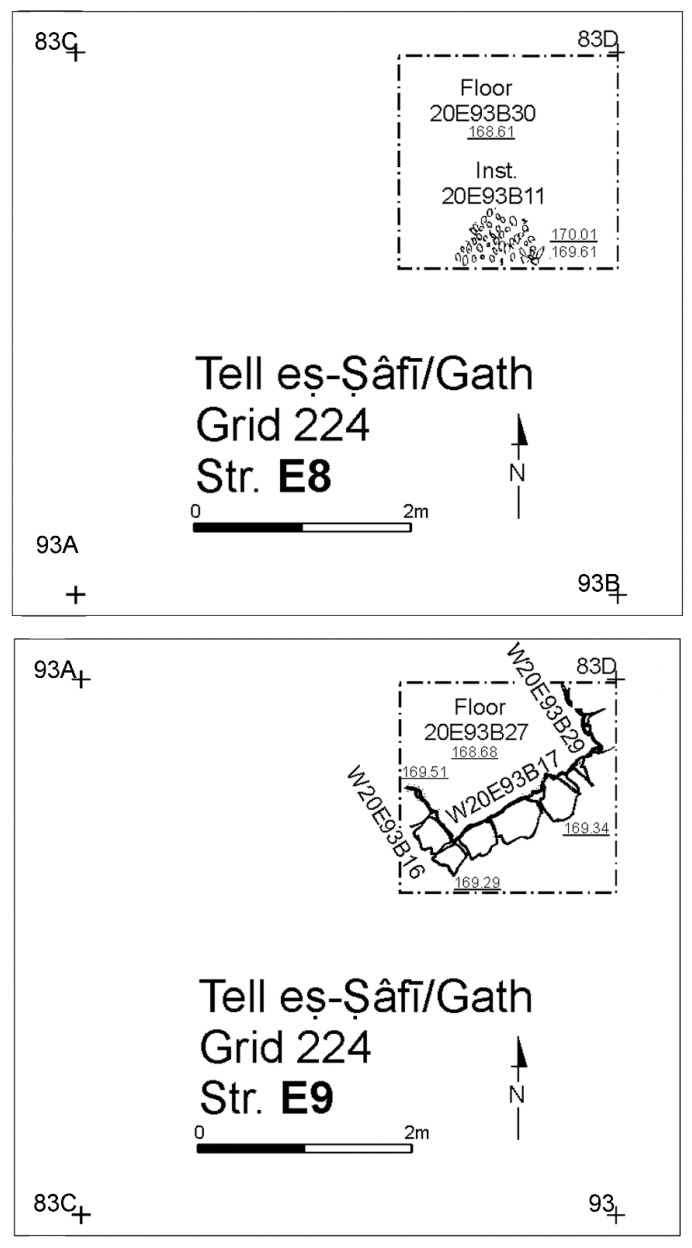
Plan of the EB III neighbourhood uncovered in Stratum E8 and E9 in the deep probe of Area E, Tell eṣ-Ṣâfi/Gath. Copyright @ Tell eṣ-Ṣâfi/Gath Archaeological Project.

There appear to be three rows of buildings which descend in elevation as one moves from northwest to southeast. In each row of buildings, the floor level declines in elevation as one moves from west (e.g., 172.70 m asl in Room 104311 of Building 104311) to east (172.18 m asl in Room 114402 of Building 114402 in Stratum E5a). This pattern is repeated in each phase of the E5 Stratum. It appears that the neighborhood is built on a series of terraces that compensates for the declining uneven natural slope of the underlying terrain. This pattern probably began in the basal E9 stratum as the floors are always horizontal in each of the ensuing EB strata. A small street or alley divides the buildings on either side into western and eastern complexes. The easternmost row probably faced onto an alley as well (Figure 11). 

### 6.2. Installations and Platforms as Evidence for Ritual

There is no evidence for niches or any special types of architectural features typically associated with ritual/cultic activity within any of the Area E buildings at Tell eṣ-Ṣâfi/Gath. However, there are features and deposits that may play a role in ritual activities. For example, Installation 94606 (in Building 134307 of Stratum E5c/in Building 74512 of Stratum E5b) is a round/circular platform (1 m diameter and reminiscent of the large stone altars at Megiddo) of solid construction built from three courses of large and roughly dressed field stones and roughly 1.5 m in height (Figure 12). It is constructed in the centre of Courtyard 114503 in Building 134307 from the E5c Stratum (Figure 7) and continues in use into Stratum E5b (Figure 6). It is replaced in Stratum E5a by a raised square platform (Installation 84109) of similar height and construction as the previous round platform (Figure 13). These installations were unique in Area E, and their precise function remains unknown. The proximity to a pebbled hearth, granary/silo, and whole vessels (related to the storage, preparation, and consumption of food) suggests that both the solid round and square platforms provided a raised work surface for everyday food preparation (among other daily subsistence tasks) carried out in the courtyard (such as those possibly linked to bread-making found at Tell Abu al-Kharaz) [81,85]. Such installations are only found within the evolutionary sequence of Building 134307 from its very beginning during the E5c Stratum to its abandonment at the conclusion of the E5a Stratum as Building 74505. It is not found in any other building or their phases.

While the precise function of such solid installations is unclear, could the round and square platforms that are consistently located in the same courtyard suggest/mark continuity in the ritual significance attached to this space? The round installation is situated near the ritual donkey burial found buried below the dirt floor of this courtyard (see below—Figure 4). Perhaps, in addition to subsistence activities, this structure served as a surface to encompass (perform?) ritual activities, in addition to subsistence.

While it is tempting to envision that these platforms were used for the ritual slaughter and preparation of the donkeys found buried beneath the floors, we recognise that this is wholly speculative and without definitive evidence.

### 6.3. Figurines as Evidence for Ritual

There are no anthropomorphic representations in the figural corpus (and iconography) from Area E at Tell eṣ-Ṣâfi/Gath. This is consistent with the general finds from residential neighbourhoods at other EB II–III sites. There are a few rider figurines from sites such as Khirbet ez-Zeraqōn and Jericho, but they are very infrequent. The rider is only schematically portrayed as the emphasis is clearly on the animal being ridden [1,37].

While most figurines at Tell eṣ-Ṣâfi/Gath are amorphous and therefore unidentifiable, except as zoomorphic, two types are recognisable: donkeys and sheep, both of which are represented in the zooarchaeological analysis [86]. The sheep and the zoomorphically ambiguous figurines are not distributed equally between all strata. Several were found in E5a (*n* = 6), one in E5b, none in E5c and E6, and two in E7. There are an additional six figurines from mixed EB stratigraphic context, though they are clearly EB because of their shape. Included among them are also fragments of donkey figurines (see below).

In the E5a Stratum, figurines are found across much of the excavation area and in a range of deposits, including walls, the alleyway, and in domestic deposits within buildings (Figure 14). Three are sheep-like, while the other three are only identifiable as zoomorphic. In Stratum E5b, an unidentifiable animal figurine fragment was found on a cobbled floor (144507) in an ancillary room (E15AQ09) of Building 74512 (Figure 15). In the E7 Stratum, the two examples were recovered from inside a wall (one sheep-like and one unidentifiable—Figure 16).

Why were so few found throughout the sequence and so many in the terminal EB occupation? Could it be a by-product of settlement abandonment, wherein the domestic space was not cleaned up beforehand, or something else (a change in ritual behaviour)?

Several of the EB figurines recovered in Area E appear to depict sheep (based on body proportions, ears, and tails, etc.). This is the most common species represented among the figurines. They closely parallel figurines recovered from the EB III round stone altar at Megiddo in the 1940’s [87]. Could this be an indication that the round and square platforms at Tell eṣ-Ṣâfi/Gath had some ritual significance? Or that a ritual (e.g., blessing) was carried out as part of the normal domestic activities taking place on the platform? While this is conjecture at this point, it is worthwhile considering that ritual blessings are a regular part of everyday life in most cultures. Many activities (e.g., food preparation and consumption) are accompanied by such blessings and one would have a depiction of the deity when saying the blessing, unless the ritual was aniconic [88].

Two figurine fragments of domestic donkeys carrying their loads were also recovered from Area E at Tell eṣ-Ṣâfi/Gath (Figure 17). One donkey figurine was found in a rubbish pit (L58040) dug deep down from the Late Bronze Age layers that disturbed the underlying EB strata. The pit contained a mixture of EB and LB ceramics. The second figurine was found on the modern surface immediately proximate to the east edge of the Area E excavations where the EB deposits rise close to the surface. Even though neither were found in secure EB depositional contexts, their decoration, shape (typology), and ceramic fabric suggest that they derive (and were disturbed) from the EB layers. They are very similar to those found at other EB sites in the southern Levant region, such as Azor, Khirbet el-Mahruq, Khirbet ez-Zeraqōn, and elsewhere in the region [1,45,89,90,91]. The presence of similar well-preserved figurines of donkeys (and other domestic animals) recovered in mortuary contexts from other EB sites within the region suggest that they are more than simple toy representations of economic animals [1].

In contrast to domestic residences, southern Levantine EB shrines have *masseboth* (sacred standing pillars/stones), altars, and other paraphernalia. However, even though there is discussion of a ‘divine couple’, the promotion of a male divinity alongside a fertility goddess, based on cylinder seal impressions [39,59] and cult processions occurring in streets [92], there is a paucity of clear anthropomorphic representations of divinity in the southern Levantine EB. Two exceptions are a cult stele that was found in EB II Arad [67] and the enigmatic ‘bearded man’/aka ‘Lord of the Desert’ stelae found on Chalcolithic and EB sites that extend from southern Jordan to Yemen [93]. The general paucity of clear depictions of EB anthropomorphic deities suggests that deities were for the most part not anthropomorphic. They were neither represented in a form that can be recognised nor made in a form that has survived. The former is unlikely given the published literature and previous developments in the Neolithic and Chalcolithic across the entire Near East and Egypt. Aniconic traditions and taboos in figural representations of deity are also a possibility [88]. Explanations that cannot be definitively ruled out also may be that the figural representations of deities were fashioned from perishable and/or recyclable materials and/or disposed offsite in a special way. This question is further addressed in the discussion section below. In any case, depictions of such deities do not appear in everyday household assemblages in the southern Levantine EB.

### 6.4. Donkey Burials as Ritual Foundation Deposits

Foundation deposits are generally found under buildings and within walls. They are buried in specific places to consecrate and dedicate the buildings to deities and to keep the inhabitants and their contents safe from harm [94,95]. A variety of animals are found in such deposits during the Bronze Age, including cattle, horses, and donkeys [96,97,98,99].

Three types of foundation deposits were recovered from the domestic buildings in Area E at Tell eṣ-Ṣâfi/Gath: votive vessels, animal figurines, and donkey burials. The animal figurines as foundation deposits were discussed already above; thus, the vessels and donkeys are discussed here.

#### 6.4.1. Votive Vessels

The votive vessel assemblage includes seven juglets, two of which are miniature objects and three are miniature bowls. All are associated with the various E5 Strata at Tell eṣ-Safi/Gath, since comparable material is found in mortuary contexts elsewhere [83].

Five foundation deposits were found associated with Stratum E5b (whole vessels). Two were found inside installations, while the other two were found inside walls. In Stratum E5a, six foundation deposits were identified, all of which were placed inside the walls of various buildings. It is important to note that, when other walls and installations from the different strata were dismantled, no artifacts were found inside the walls; hence, the suggestion that the whole vessels found in walls and installations were foundation deposits.

Foundation deposits are also known in Egypt from the Early Dynastic and Old Kingdom onwards. In Egypt, the placing of a foundation deposit was part of the initiation ritual of a building and reflected the construction process. Due to the similarities between Egyptian and Canaanite ritual ceremonies, it is possible that there was an Egyptian influence in the adoption of foundation deposits during the Early Bronze Age in Canaan [100,101,102].

#### 6.4.2. Donkey Burials

The donkey burials from Area E are outlined along with their significance as foundation deposits. In the E5 stratum and its various phases, seven completely articulated domestic donkeys and one partially disarticulated skeleton were excavated from their shallow pits located beneath the floors of the buildings (see Figure 5, Figure 6 and Figure 7). The frequency of burials is not consistent across the various E5 strata. There are four in the earliest stratum (E5c) when the neighbourhood of buildings is constructed (Figure 7). Subsequently, two additional burials were found within each of the strata (E5a and E5b) when the buildings were renovated (Figure 5 and Figure 6) [103,104]. In every instance, there was no evidence of post-mortem disturbances or re-orientation of the skeleton, such as rodents, gnawing, weathering from re-exposure, burning, etc. that would have damaged and/or moved the skeleton. Nor is there any evidence of other objects placed within the pits containing the donkey internments.

In Stratum E5c, four donkey skeletons were found in buildings on either side of the narrow alleyway (Figure 7) (Note: Donkeys 1–4 from Tell eṣ-Ṣâfi/Gath have been previously discussed in other publications (e.g., [103,104] These data are summarised and integrated here with previously unreported data on several of the other donkeys.). Donkey 1 (L114506) was found in a shallow pit below the floor of Courtyard 114503 of Building 13407 to the east of the alleyway (L20E93A02—Figure 18). It is located at the south side of the courtyard alongside and below the foundations of the wall of the building (W75611).

This donkey burial is the clearest example of ritual internment [103,104]. The skeleton was carefully placed in the pit on its right side with the torso facing west (toward the setting sun); the front and hind legs were tied together (trussed) below the abdomen, and the upper neck (cervical) vertebra and cranium dismembered and placed on the abdomen facing east (toward the rising sun) (Figure 19). There was no evidence of any other objects found associated with the burial. It is evident the animal was sacrificed, the head fully cut off, and carefully placed on the abdomen facing in the opposite direction [103]. Both front legs are bent backwards at the humerus–radius/joint, and the hind legs are pushed forward so that they meet below the torso and parallel to the vertebral column. The only way that the legs could have been placed in this position, which is contrary to their natural state, would be if the legs were bound/trussed/tied together. It is likely that the donkeys were carried here by a pole through its tied legs. However, no evidence of rope remains were discovered in any of the burials. The skeleton was not buried in a haphazard manner by being dumped into the pit, nor did it fall into a pit naturally. It was slaughtered, carefully laid in the pit on its side, and the dismembered head laid on the stomach facing towards the rising sun before being covered up.

Donkey 1 (L114506) differed from all subsequent burials in Area E with regards to the treatment and placement of the head in relation to the rest of the body. While most of the skeleton is in normal articulation, the neck (cervical vertebrae) and head skeletal elements (cranium and mandible) are not in their normal position. Neither can their position be accounted for as a result of natural decomposition. The first and obvious fact is that the head and cervical vertebra are in reverse anatomical position in relation to the rest of the skeleton. These elements appear to be disarticulated from the remainder of the skeleton and are facing the opposite direction (Figure 19). The neck and head were carefully detached (severed) from the rest of the backbone (thoracic vertebrae) and placed on top of the stomach/ribcage prior to burial. There is a space between the cervical and thoracic vertebrae, implying that the head and neck were dismembered from the rest of the skeleton. This anatomical position is highly unusual and cannot be explained as a result of natural death, being thrown or falling into the pit, etc.

**Figure 19 animals-12-01931-f019:**
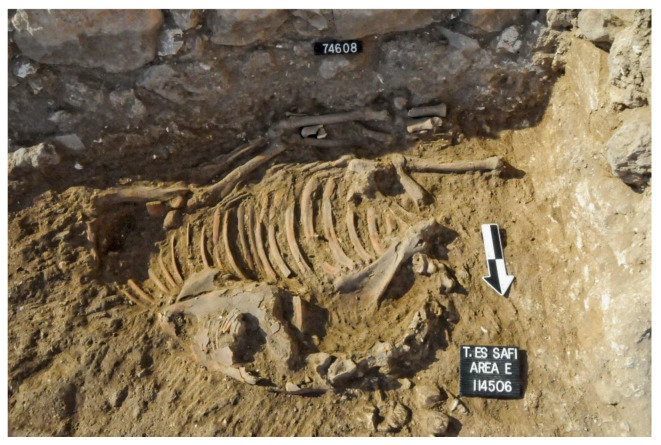
Photograph of Donkey 1 (L114506) found in a shallow pit below the floor of Courtyard 114503 of Building 13407 in Stratum E5c of Area E, Tell eṣ-Ṣâfi/Gath. Copyright @ Tell eṣ-Ṣâfi/Gath Archaeological Project.

The animal was likely slaughtered by cutting the throat. Ultimately, the muscles that hold the head in the anatomically correct orientation were severed to separate the cranium and upper neck (cervical vertebrae) from the rest of the torso. It is impossible to twist the neck of an animal this large into such a position and to maintain it afterwards (even if broken) without severing the muscles first. It would simply bounce back into its proper anatomical position. The neck must have been slowly and carefully severed after the animal was already dead, since there is no evidence for chop marks on the bones. The head and neck were likely severed slowly by knives slicing carefully rather than a rapid chopping action that would have damaged the bones and been more obvious.

The depositional context, orientation, and layout of Donkey 1 provides the clearest and most unambiguous evidence it was a sacrifice. It is clear from the carefully laid out anatomical orientation of the donkey skeleton that it did not fall into the pit while alive. Similarly, it was not thrown into the pit after death. It was purposely and carefully laid in the pit after death by slaughtering and dismemberment. The position of the legs informs us that it was bound at the hocks (carpal/tarsal joint), most likely before it was placed in the pit. It likely died from a flesh wound, such as the cutting of the jugular of the neck, to allow the blood to seep out so that it could be used in any associated rituals. As will be shown below, the heads of all three donkey burials from the ‘House of Asses’ also faced east even though they were not dismembered.

The other three donkey skeletons from Stratum E5c (Donkeys 2–4: L19D82D04, L19D83C09, and L20D93A05, respectively) were found in similar shallow pits buried below the floor of Courtyard 17E82D02 in Building 17E82D08 (west of the alleyway) (Figure 20). Building 17E82D08 was nicknamed ‘the House of Asses’ as a result. Given the nature of buildings in this area (courtyard and small adjacent room), only the eastern half of this building was exposed. The western half of this building was not excavated, as it is sealed by several metres of Bronze and Iron Age deposit/overburden. It cannot be certain if there were more ass burials to the west, as it is sealed by a thick overburden (5–6 m) to the west of Area E and must await a future generation of excavation.

The skeletal orientation of the three donkeys in the “House of Asses” differs from that of Donkey 1 (L114506) across the alleyway in some significant ways. First, none of the Donkey 2–4 heads were dismembered, contrary to Donkey 1. Second, their complete skeletons were laid out on their left sides while facing the rising sun. In contrast, only the head of Donkey 1 was laid out in such a manner, while the rest of the body was laid out on its right side and facing toward the west. Third, the fore and hind limbs of both sets of donkeys were in unnatural positions. The limbs of Donkey 1 were pulled tightly under the torso so that they overlapped. In contrast, the fore and hind limbs of Donkeys 2–3 were less bent and did not overlap. They appear to be bent only just enough at the elbow and knee joints to fit into the shallow pit.

There are also similarities with respect to all four donkey burials beneath buildings on either side of the alleyway that provide evidence for a common mode of treatment during burial. First, the cranial orientation for each of the donkey skeletons was facing east towards the rising sun. Second, while the head of Donkey 1 was clearly dismembered, none of the donkeys from the House of Asses exhibited slicing marks (Note: In a previous publication, it was mentioned that there was a butchering mark on the epistropheus (2nd cervical vertebra) of one of the “House of Asses” donkeys—Donkey 4—L20D93A05 [105]. However, recent closer microscopic examination of modern donkey specimens curated at Hebrew University and Tel Aviv University demonstrated that the so-called butchering groove was in fact a natural morphological feature commonly found on this osteological element.). Third, they were all buried in shallow pits below the dirt floors of the courtyards before the walls and floors of the buildings were constructed. It is assumed these spaces were open courtyards due to the size of the rooms, the general absence of pillar bases (c. 8 × 8 m), and the presence of adjacent narrow rooms, often with pebbled floors. Crucially, there is no evidence or indication for pits that would have cut down through the Stratum E5c floors when the rooms were subsequently occupied. In other words, the burials are not intrusive from the overlying occupational levels.

Each of the Stratum E5c donkey skeletal deposits are securely dated to the construction phase of the Stratum E5c buildings—in other words to the archaeological “moment” when the neighbourhood is being renewed. They are slaughtered as sacrifices in the time between the demolition of the Stratum E6 buildings and before the construction of the Stratum E5c buildings. All the burials stratigraphically predate the construction of the overlying E5c architectural features and floors that seal the donkey skeletons and thus represent foundation deposits.

The pattern of donkey burials is repeated in the E5b and E5a Strata but in modified form. Two donkey burials were found in Stratum E5b—Donkeys 5 and 8. The former is complete, and the latter is a partial skeleton. Donkey 5 (L144511) is a complete skeleton of an infant found in a pit below a Stratum E5b hearth/cooking installation in Room E15AQ09 (Figure 6 and Figure 21) in Building 74512 on the east side of the alleyway (L19E83C12). The inhabitants used this part of the room/building for cooking throughout the entirety of the E5 Strata. The infant donkey was found in a pit directly below and sealed by the E5b installation. Therefore, the internment of the donkey can be stratigraphically dated to when the room was renovated/renewed. It appears that the cranium of the infant donkey was oriented toward the east in a direction similar to those in the E5c Stratum.

Donkey 8 (L17E82D09) is a partial skeleton found in a small pit beneath the dirt floor in Courtyard 16E82D05 in Building 1682D05 on the west side of the alleyway (L19E83C12) (Figure 22a). It is composed of several disarticulated osteological elements (a cranium/maxilla and cheek teeth, scapula, and humerus) from a partial donkey burial. No special orientation could be discerned. Based on the maxillary tooth eruption and wear sequence, it belongs to a young juvenile (6–8 months old) since the upper dm2–4 are well-worn, while the UM1 is almost fully erupted but not worn (Figure 22b). It is interesting to observe that the two E5b donkeys were very young animals, contrary to the Stratum E5c specimens.

Stratum E5a produced evidence for two additional donkey burials. The first (Donkey 6—L74517) was a partial skeleton buried in a pit beneath the floor of Courtyard 94413 of Building 94413 (Figure 5 and Figure 23). It includes most of the skeleton. Most of the thorax (ribs, costal cartilage, and vertebrae), forelimb (scapula, humerus, carpals, and phalanges) and hindlimb (innominate, tibia, tarsals, and phalanges) were also present. While a loose tooth from the mandible (LM1) was found, the rest of the cranium appears to be missing. This individual was mistakenly published as a pig in the first volume of the excavation reports on the site [106]. While the overall orientation of the skeleton appears to be from north to south, it is impossible to reconstruct the cranial orientation given its absence. It belongs to an old subadult given the state of epiphyseal fusion on the long bones and vertebrae.

The second Stratum E5a donkey burial (Donkey 15—L144506) was that of an older infant. It was found poking out of the 5–6m high west baulk of the excavation area where it meets Wall 104206 of Building 104311 (Figure 5 and Figure 24). Only the distal (lower end) of the legs (metapodia, carpals/tarsals, and phalanges) of three limbs were found. They are in articulation which suggests that a complete donkey lies further in the baulk. The legs extended slightly from the baulk into the alleyway (104306). The orientation of the legs suggests that the torso of the animal would be beneath Wall 104206 of Building 104311 (or more likely, the joint wall of the adjacent building, given the size of the rooms), as the wall of the building appears to intersect where the torso would be. The rest of the donkey awaits excavation by a future generation. Given the orientation of the legs, the donkey skeleton would be in a northeast–southwest orientation, and the head would most likely have pointed east toward the rising sun. This suggestion is purely conjectural at this point, however.

In general, three of the four donkeys from the renovation strata (Strata E5a and b) are of different age classes (infant—0–6 months) than the donkey deposits associated with the construction of the buildings in Stratum E5c (old subadults/2.5–3 years old or young adults/3+ years) in this neighbourhood. This could signal that household rituals are characterised by greater variability, hence flexibility.

Earlier phases of the EB III neighbourhood (Strata E6–E9) did not produce additional donkey burials (Figure 8, Figure 9 and Figure 10). However, this is not certain, as the size of the excavation area in the underlying strata become progressively smaller than as one moves down in the stratigraphic sequence from E5 to E9 Strata. Most of the floor deposits were excavated for the E6 Stratum, but exposures were far more limited for the underlying phases. In these earlier levels, only scattered elements of donkeys were found disbursed/distributed amongst the food debris in the accumulations on/above the floors. It is plausible that more donkey burials are present, but a firm conclusion must await further excavation.

In total, the E5 Strata from Area E produced eight complete and partial donkey skeletons buried in shallow pits dug into the layer created by the demolition of the E6 mudbrick buildings. The Stratum E6 building was pulled down, and the mudbrick superstructure was packed down before the construction of new stone foundations and new dirt floors for the next stratum of occupation (Stratum E5c). The donkeys were intentionally deposited by the occupants of the buildings. Only afterwards were the floors constructed, thus sealing the donkey burial foundation deposits under the courtyards of the building. The donkey burials in Strata E5b and E5a were similarly buried as the buildings were being renovated. It appears that, with each major construction or renovation stratum, donkeys were slaughtered and buried in shallow pits beneath their house floors.

Only one of the donkey burials had any clear evidence for slaughter. The head and upper neck of Donkey 1 (L114506) from Stratum E5c was dismembered from the rest of the body and placed on the thorax. There is no evidence that any of the donkeys were otherwise dismembered, cooked, and/or eaten. Furthermore, no ceramics or other artefacts were placed in the pits.

Some clear and consistent patterns emerge with regards to the deposition of all the donkey skeletons throughout the E5 Strata. First, where it was possible to identify, all the Strata E5c, one E5b, and one E5a skeleton were either entirely oriented or only the cranium was pointed toward the rising sun. Second, the age classes shift from the construction to the renovation phases as well—more older subadults/young adults during the construction of the E5c Stratum, while more younger animals (infants and juveniles) were buried during the Strata E5a and E5b renovations. All of the sexable individuals were female. Third, in terms of layout, all the donkeys were placed in shallow pits immediately below the dirt floors associated with the deposit. The legs of each were carefully folded to fit within the shallow ellipsoid shaped pit. The four E5c donkey skeletons yielded the best information on the special nature of the donkey deposits in this regard. Fourth, based on the shape, depth, and orientation of the burial pits and their relationship to the surrounding deposits, the donkeys were interred when the buildings were renovated and the architecture renewed, but they predate the occupation of the buildings in each of the phases. The timing is important because the deposits are not attributable to the ‘daily life’ of the inhabitants when the rooms and courtyards were lived in/occupied. This burial context is a key reason why we regard these burials as ‘foundation deposits’ interred during construction/renovation and not simply rapid ‘animal disposal’ during the subsequent use of the buildings.

The pattern of donkey burials in this part of the site (Area E) is repeated through time and thus qualifies as a habitual commemorative practice marking the foundation and renewal of buildings (not conditioned by material/functional constraints). When buildings were renewed and remodelled, the old buildings were collapsed and packed down, before new walls were laid down on earlier wall stubs. Complete or partial donkeys were sacrificed and buried in the interim before the floors were installed in each stratum of E5. Therefore, not only was the construction of these buildings/dwellings consistently and carefully planned (organised and built), but the entire process was embedded (sanctioned/punctuated) in ritual. Thus, the boundaries between the sacred and profane are blurred, which reinforce each other. Arguably, these deposits signal a process of ‘commemorative history making’ (see discussion above) that consolidate long-term alliances/ties and social networks. These not only would have sustained the immediate household in the present [97], but also made possible its reproduction over time. Thus, they would have contributed to the larger social and economic configurations distinctive of the early urban lifestyles and culture in the southern Levantine EB (thus making possible the intensification of agriculture, trade, and exchange, a landscape of fortified tells and so forth in economies increasingly characterised by delayed returns for high-labour investments). The high number of donkey burials suggests community-oriented ritual activity (at the scale of a neighbourhood/cluster of houses?) akin to so-called ‘history houses’ [33]. This may explain why only some buildings produced evidence of donkey burials and solid platforms in each of the strata.

## 7. Textual Analogies for Donkey Ritual Internments

While there is only clear evidence with regards to the mode of death for one of the donkey burials from Area E, there is indirect evidence for the selection of certain donkeys to be part of these ritual internments associated with the construction of the neighbourhood. First, there is their depositional context—all four E5c burials in shallow pits beneath the floors of buildings about to be constructed. Second, the sex and age-at-death of each of the four E5c donkeys is remarkably similar. All are relatively healthy females, either young adults or older subadults. These are “expensive” and valuable animals given that they were just at the age when they would be carrying loads and becoming sexually mature. In fact, they are a form of wealth that is being taken out of circulation, as they are slaughtered prematurely as part of the rituals associated with the rebuilding of the neighbourhood in Stratum E5c. The chance that four donkeys of similar age and sex were buried in similar ways in courtyards of domestic buildings on both sides of the alleyway at the same time when the neighbourhood was being rebuilt seems remote and unlucky to say the least (aside from the huge economic loss this would inflict). Moreover, even a quick perusal of ancient Near Eastern or Egyptian literature demonstrates that one does not present to the gods an offering of sick, injured, or diseased animals. Thus, the burial as the neighbourhood is being rebuilt of several healthy females facing the rising sun and just beginning their reproductive years is in our opinion a smoking gun.

However, one cannot use the analogy from the Hebrew Bible (Old Testament) or slightly later cuneiform sources to reconstruct the nature of the sacrifice of a young adult donkey. In both the Iron Age Biblical traditions and MB Mari cuneiform documents, wherein donkey sacrifice is mentioned, there is an emphasis upon foals. In the Hebrew Bible, the first born must be redeemed or its neck broken (Exodus 13:13). At Mari, foals are also clearly specified:
*“They brought me a puppy and a hazii-bird to ‘kill’ the donkey foal (i.e., make peace) between the Haneans and Idamaras but I feared my lord and did not give over the puppy and hazu-bird. I had a donkey foal whose mother was a she-donkey killed (and) I established peace between the Haneans and Idamaras”.*(ARM 2 37:6-14) [107]

Furthermore, it is not possible to use analogies with the Hebrew Bible since males are the animal of choice for sacrifices, and the Tell eṣ-Ṣâfi/Gath donkey skeleton is a female.


*“And every firstling [male in Hebrew] of an ass you shall redeem with a lamb; and if you will not redeem it, then you shall break his neck…”.*
(Exodus 13:13)

Donkeys are the only non-kosher animals that are used in a sacrificial context. Sacrifices are linked to the tradition of redemption of the first-born male donkey [97,108,109,110]. Clearly, the Tell eṣ-Ṣâfi/Gath donkey skeleton does not fall into these categories. Additionally, the EB III long predates Biblical traditions, making any analogy difficult.

It is interesting to note that donkeys are generally female in most ancient literary sources wherein the gender is noted. This is similar to the choice of sex for the EB donkeys at Tell eṣ-Ṣâfi/Gath. It is possible to attribute this choice to donkey ethology, as female donkeys tend to be more amenable than uncastrated male donkeys, who do not get along well in mixed-gender herds because of issues of behavioural dominance. Further, donkeys in general are highly territorial, and females are known for being very protective of individuals in their social group. Modern farmers still use female donkeys to protect their herds in areas where predators are prevalent. Perhaps this quality was recognised and is an additional reason why mature females were always chosen as foundation deposits. However, we feel this is unlikely given the emphasis upon females just reaching their reproductive years. Ancient Near Eastern religions prize animals that are not only healthy but also that signify fertility [107,109,111,112,113].

In general, the Tell eṣ-Ṣâfi/Gath donkey skeletons belong to young and healthy animals. They do not display any evidence that the animal was sick or suffered from any major osteological injuries or deformities. For example, there are no broken and or healed bones, dental abscesses, dental malocclusion, bone lesions, or severely arthritic joints. This corresponds with the preferences for ritual sacrifices in ancient Near Eastern religions for healthy (and conscious) animals. A healthy animal is needed to appease or placate the gods and to sanctify agreements [107,114].

However, two types of pathologies were eventually discovered on the donkeys that have implications for how the animal was used during the EB. These pathologies illustrate evidence for bit wear on teeth (for riding) and foot malformations (distal limb) characteristic of carrying loads over uneven ground [91,115]. Both would be relevant for animals used as pack animals, particularly in caravans over difficult terrain. The evidence from the dental isotopes supports this conclusion, as some of the donkeys and goods found at the site come from great distances, such as Egypt [44,116].

## 8. Discussion—Why Bury a Domestic Ass under Your Floor?

The increasing frequency of domestic asses coincides with the dramatic rise in trade across the region beginning with the EB. Over time, asses play an important role in regional and inter-regional exchange and transportation, as goods are moved across all kinds of terrain. During this period, there is an increase in the quantities of goods, such as copper and other products (e.g., Canaanean blades), being transported across and between regions [45,46,117,118].

A donkey provides a mechanism to accrue and distribute wealth (whether ‘staple wealth’ in the form of agricultural products or ‘finance wealth’ in the form of precious exotics and metals with high exchange value etc.). Those with donkeys presumably had far greater potential to be ‘socially mobile’ than those without. The use of the domestic ass (*Equus asinus*) to transport goods across and between regions would have allowed a change in the scale of economic systems. In contrast to the use of human carriers, this would have lowered the costs of goods. Further, the larger quantities of goods that a train of asses could carry with a single driver would generate economies of scale. The owners of domestic asses would directly benefit. A consequence was the emergence of a new social class—i.e., merchants who had asses/donkeys that could transport goods over near and long distances [1,45,46]. Their asses would be the source of their growing wealth and power.

### 8.1. Donkey Caravaners and a Specialised Merchant Class

A class of specialised merchants, such as ‘donkey caravaners’, probably existed by the beginning of EB I in the southern Levant. They would have specialised in the transportation of commodities. Donkeys, as with any large domestic animal, take years of investment in raising and feeding before they can be used for their secondary products—pulling/draught and carrying/transport. It is possible that such merchants began to exist toward the end of the Chalcolithic, since there are examples of pre-EB laden animal figurines [90,119,120]. The appearance of EB laden animal figures was probably closely associated with spread of the domesticated ass throughout the Near East. At Tell eṣ-Ṣâfi/Gath, two fragments of laden donkey figurines were recovered from Area E (although in poor chronological context—see above). However, they belong to a very common type attested throughout the region during EB III on typological grounds [91].

Further, both fragments show the panniers attached to a donkey (Figure 17). The donkey itself is mostly missing, but the straps to stabilise the loads/baskets are illustrated in red paint, and as applied decoration. If loads/weights are evenly and securely distributed, donkeys can carry up to 20–25% of their own body weight. Such knowledge appears to have been reflected in how donkeys were routinely depicted in the figurine corpus/figural world (and explains the pathologies identified on the skeletons of the burials at Tell eṣ-Ṣâfi/Gath) [91]. Similar donkey figurines are widely known/reported from other sites and are found in a variety of contexts, especially mortuary/funerary deposits (e.g., Tombs 20 and 60 at Azor) [1].

Merchants and/or ass herders would have occupied specialised positions and roles in the increasingly complex social structure of Near Eastern state and urban societies (as evidenced by the extensive EB Mesopotamian [121] and MB (Old Assyrian) archives [122,123]). As with all guilds or classes, there will be rituals or ceremonies associated with their activities. While the ass was a means of transportation, it was also an important element in ritual and would represent a totem with an associated cult for groups who rely upon them for their livelihood, i.e., merchants and/or donkey herders [45].

### 8.2. Sacred vs. Profane—Butchered Donkeys

Donkeys were an essential element in the system of transportation that moved goods within and between regions. Donkeys are expensive to have and maintain. Contemporary studies of donkey exploitation in the developing economies of the Third World, such as exists across much of Sub-Saharan Africa, demonstrate that household ownership of a donkey is a reflection of greater status and prosperity compared to households without a donkey [124,125,126]. Donkeys are primarily used as a transportation animal for people and to transport crops from the fields and goods to market. It was thought that donkeys were not consumed and were never a major food source, given their relatively low frequency in the zooarchaeological remains from this and other EB sites [1,86]. Milevski and Horwitz (2019) go as far to suggest a dietary taboo on donkey meat was prevalent in EB society. This was clearly not the case at Tell eṣ-Ṣâfi/Gath in EB III, where, in addition to articulated donkeys in burials below floors, there are loose donkey bones mixed with food debris on and above the floors (at c. 3%). Several of these bones exhibit butchering marks, in clear contrast with the articulated donkeys found in burials below the floors (Figure 25) [105]. In such situations, there is a demarcated boundary with a dichotomy between donkeys chosen for inclusion as part of ritual foundation deposits versus donkeys exploited for food and whose remains are found mixed with the remnants of other animals on the menu. In other words, there are differences in context, deposition, treatment of the bones, age-at-death, sex, etc. between those used for sacred and profane activities. No other animal in the faunal assemblage is characterised by this dichotomous split.

### 8.3. Neighbouring Parallels and the ‘Cult of the Beast of Burden’

Complete burials of young female donkeys in pits below floors are known from several EB sites across the Shephelah (coastal foothills) and neighbouring coastal plain (e.g., Azekah, es-Sakan, Nahal HaBesor, and Lod). At Tel Azekah, two decapitated infant donkeys were buried in a shallow pit below a slumped EB III floor [127]. The Tel Azekah donkey were laid on their right side with the head turned around to face the tail. The tail of each faced in opposite directions—Donkey 1 was oriented south (fore) to north (hind), while Donkey 2 was oriented to the reverse. While the position of the Tel Azekah Donkeys 1 and 2 crania are reminiscent of the Tell eṣ-Ṣâfi/Gath Donkey 1 (pointing towards the tail), they faced very different directions. The cranium of Donkey 1 from Tell eṣ-Ṣâfi/Gath pointed toward the east, while that of the two donkeys from Tel Azekah faced toward the north and south, respectively. As with the Tell eṣ-Ṣâfi/Gath E5c donkey burials (Donkeys 2–4), the hind and forelegs of the Tel Azekah donkeys were similarly positioned in a way that suggests the pairs of legs were bound together but not front to back. Only the Tell eṣ-Ṣâfi/Gath Donkey 1 burial displays a pattern that suggests that front and hind legs were bound together. Sites with high donkey bone counts in EB deposits are many (e.g., Lachish, Jericho, EB I Ashkelon, the Halif Terrace/Nahal Tillah), but it is not always clear whether they originated from intentional burials in pits below structures. For a comprehensive survey, see [1].

As can be illustrated from the above data, domestic donkeys begin to be sacrificed and buried beneath the foundations of non-elite residences during the EB and are not simply beasts of burden. They also have a holy or symbolic role to play among the families whose homes they are buried beneath. These donkey deposits are ritualised burials and their use as foundation deposits suggests that they were intended to sanctify the household (both the physical structure and the well-being of the kin inhabiting these structures). Milevski has long advocated for a “cult of beasts of burden” during this time period [45]. Such a cult occurs in segments of societies that rely upon asses for their livelihood. Such practices are similar to those found elsewhere in the world, such as in the Andes, where beasts of burden are associated with feasting (and other forms of social gathering) and invocations of supernatural powers (such as to protect the owners and their animals and to increase the prosperity of the owners) [1,45,46,128]. This is not as far-fetched a proposal as one may suppose. As Mitchell notes:
*“Within the Ancient Near East, three themes stand out: the donkey’s role as an indispensable vehicle for moving goods over both short and long distances, most conspicuously in the trade of metal and textiles between Assyria and Anatolia in the early second millennium bc; its elite associations as a prized riding animal; and its religious significance as reflected in rituals governing the conclusion of treaties, the celebration of festivals linked to individual gods, and the curing of illness.”.*(p. 11 of [129])

The iconography of asses in the EB (in the southern Levant) is limited to a narrow range of motifs that are found across much of the region. Milevski interprets this to signify not only the role of the animal in daily life but also that the animal was a symbol for social groups that relied on asses for their livelihood, such as merchants [45,46,118]. Asses as totems represent commerce (trade more specifically or exchange more generally) and do not represent simple obstinacy as they do in modern Western-oriented cultures.

The discovery of equid (asinine) burials with evidence for special treatment beneath the floor of houses is unusual but not without precedence, given the evidence for ritualised donkey sacrifice at other domestic and nondomestic sites in the southern Levant and Egypt during the 3rd and 2nd millennia BCE [97,98,99,103,104,130,131,132,133,134,135]. This location is usually reserved for sacred deposits. Sacrificial animals are buried as foundation deposits to appease the gods and to sanctify and protect the occupants [131]. It is therefore not a coincidence that such animals were singled out for such special roles [136], especially given its known importance in later texts (as a purification sacrifice in Canaanite liturgy at Ugarit, for treaties at Mari, etc.). However, caution is necessary in projecting information from later texts onto the archaeology of earlier periods, especially given the enormous time gap (more than half a millennium) (Mari, i.e., ARM 2.37:11; A.1056:9–10; A.2226:17, 15) [107,137,138].

The ritual and symbolism surrounding domestic asses during the EB have both elite and private origins. EB ass burials (including hybrids) are not only associated with elite burial and public contexts [97,139,140], they are also associated with commoner residences where they are buried beneath floors and walls [1]. We believe that the asinine burials from the E5 Strata at Tell eṣ-Ṣâfi/Gath are such examples and fall into this category.

## 9. Conclusions

It is possible to pose several observations about the nature of domestic rituals in an urban settlement, such as EB Tell eṣ-Ṣâfi/Gath. First, household rituals are very personal, as they are related to the immediate surrounding world. Second, household rituals are implanted into the base fabric of their homes as ritual sacrifices of valuable animals and creations of votive items, such as animal figures, planted into the floors and walls of homes. Third, household rituals are intimately related to lifestyles—to what is important. The occupants are choosing animals that are important to their lifestyle and economy—shepherding of sheep and transporting goods and people with donkeys. Fourth, household rituals/cultic behaviour are not for the general public. They are private and intimate and are unlikely to be formalised. Fifth, household rituals are not governed by the more public rituals of temple and other elite institutions. Their diversity suggests that they manifest a religious independence or freedom.

The EB foundation deposits at Tell eṣ-Ṣâfi/Gath and elsewhere (across the ancient Near East) appear to be part of a very long and enduring tradition that reveres donkeys. The veneration of donkeys is as old as the institution of the state itself and urbanism. This is apparent given the presence of donkey skeletal material by the Late Chalcolithic Mesopotamia and EB I in the southern Levant, as well as their association with the burial rituals of the earliest pharaohs. Even these dates may be too late, since there appears to be a depiction of a domestic donkey on a ceramic sherd from the 5th-millennium BC site of Tol-e Nurabad in Western Fars, Iran [141,142]. As such, the spread of the donkey appears to coincide with the rise of social complexity across the Near East.

The donkey skeleton burials excavated in the later EB stratum of Area E at Tell eṣ-Ṣâfi/Gath were found in shallow pits sealed beneath the floors in each building stratum, shortly before the dwellings were occupied. The animals were deliberately bound, possibly slaughtered, and buried as a foundation deposit to bless the construction of houses as the EB III neighbourhood was rebuilt. In most cases (only one exception), the skeletons belonged to young and healthy late subadult/early adult females.

The choice of a donkey for such a ritual activity implies that it was an important religious symbol for the occupants of the neighbourhood. Given the distribution of these donkeys, on both sides of the alley (and under/in courtyards), perhaps the constellation of these deposits can be used as an indicator of the number of households in this area (as opposed to calculating the number of households from the number of buildings/physical structures). Transcending physical structures is obviously difficult, but evidence for ritual activity could be a useful indicator in the study of household behaviour.

The asinine burials from Area E at Tell eṣ-Ṣâfi/Gath are usually buried under courtyards (except for an infant donkey buried from Stratum E5b buried in the corner of a room in a hearth after it has gone out of use—it is rapidly buried beneath the new floor of the room) [104]. This suggests purposeful selection for this location, which does not appear to be a coincidence. Courtyards are the locus of most household activities. It is the focal point of the activities in the building—food preparation, consumption, tool making, and other activities. Based on this information, burial location and treatment (of complete skeletons—since bones are not often found in articulation in a neighbourhood and since the skeletal orientation implies special treatment) is suggestive that these asinine burials represent ritualised deposits and not simply ‘donkey disposals’. No other animal receives comparable treatment in the entire faunal assemblage at Tell eṣ-Ṣâfi/Gath or other EB sites in the southern Levant.

The presence of items of non-local origins from the EB stratum in Area E at Tell eṣ-Ṣâfi/Gath, such as an ivory cylinder seal [143] and other exotic and quotidian items, such as alabaster mace head, faience beads [73,144], and ground stone objects [145,146,147,148], alongside the donkey burials, suggests that the dwellers of this domestic neighbourhood were not from the lowest rungs of the socio-economic ladder but rather belonged to an emergent/evolving merchant class [44]. The household assemblages from Area E reflect considerable mobility—a mix of varied local and long-distance trade and the exchange of ‘quotidian’ and ‘exotic’ objects used and needed in daily life. The EB III residents in Area E had sufficient wealth and/or access to sacrifice an expensive animal, a young female donkey. The residents of the Tell eṣ-Ṣâfi/Gath ‘House of Asses’ commanded sufficient resources to lose at least three female donkeys.

It has been suggested here, and elsewhere, that these donkeys were slaughtered and buried as a foundation deposits because the inhabitants of Area E may have been merchants, whose totem was a donkey [103]. The two zoomorphic vessels/donkey figurines from this part of the site described in this paper strengthen the linkage of the donkey with merchant behaviour. The presence of minor osteological pathologies related to injuries from carrying heavy loads over uneven ground further supports our interpretation that this is a neighbourhood of merchants [91]. In addition, evidence of dental wear on the LPM2 suggests that a soft material bit was used on the donkey, possibly for riding. (Figure 26). It is possible that this young adult female animal was used in a special role, such as for riding, and not for simply carrying heavy loads. This would explain both the bit wear and the presence of low-level osteological pathologies on the lower rear extremities. Hence, the use of such animals for riding was not limited to the elite.

The evidence from Tell eṣ-Ṣâfi/Gath and other EB sites across the Near East demonstrates that donkeys figured prominently in elite and commoner, and the public and private rituals of the peoples of the southern Levant and probably across much of the Near East. The sacrifice and burial of donkeys is most likely part of a long tradition that extends from at least from the Chalcolithic and Early Dynastic Mesopotamia across the Near East to the Old Kingdom in Egypt. As the evidence presented here demonstrates, the importance of domestic donkeys extends beyond the economic into religious realms.

In keeping with the above, it is suggested that the donkey was a religious symbol for a specific demographic social stratum and occupational group (merchant caravaners/guilds). The inhabitants of the Area E neighbourhood at Tell eṣ-Ṣâfi/Gath appear to fit somewhere between traditional definitions of “elite” and “nonelite”. As noted elsewhere, there should be the recognition of a social stratum intermediate between the simplistic dichotomy of elite and nonelite, given the textual evidence of social complexity by this time in neighbouring ancient Near East societies—there are “special positions or roles for different segments of society” [127]. Thus, the excavated domestic residences of people with non-elite status at Tell eṣ-Ṣâfi/Gath suggest that these were residences of social groups with unique and specialised status. They were wealthy enough to afford to sacrifice valuable animals—i.e., merchants. Thus, donkey burials beneath the floors of such buildings may signify the presence of the homes of merchant families. They appear to be located mostly on the periphery of urban settlements. A recent reanalysis of faunal remains from other EB sites in the Near East has also suggested something similar [149]. Therefore, such donkey burials are a previously unrecognised, but important, archaeological diagnostic of this emerging social group that has been traditionally overlooked by most archaeologists. Travel on trade routes was highly dangerous in most/all periods, and the nomadic lifestyle can be brutal and highly demanding, hence the need for supernatural/divine protection.

The many discoveries across the Near East show that the practice of sacrificing donkeys was not an isolated and unusual occurrence. They are foundation deposits clearly linked with the physical renewal of dwellings. The deposits were presumably interred by prospering households that repeated this tradition over time (strong continuity). Our research suggests that this consistent pattern signals the presence of merchant guilds/caravaners with specialist knowledge (a distinct social stratum), who were located in this part of the city during the EB III of Tell eṣ-Ṣâfi/Gath.

The donkey was not necessarily the focus of worship for the entire population. The sacred landscape was presumably highly heterogeneous/plural (multi-layered) with different social groups or ‘communities of practice’ attached to, and associated with, specific deities and ritual activities. For example, in the “Temple of the Serpents” at Jebel Mulawwaq in Jordan [61,150], the pottery was decorated with many depictions of snakes and trees (the trees also resemble snakes). Distinctive horned bulls decorate many of the slabs in the sacred ‘picture pavement’ of Stratum XIX at Megiddo (Slabs 1, 8, 14–18, 23) and are thought to be symbolic of Egyptian-style royal power [151]. Even though these examples are from the EB IB (not EB III), when there is a richer iconography related to cultic activities and temples, it is a reflection of what might be missing in the EB III archaeological record. These ‘communities of practice’ would occasionally unify/converge under the umbrella of a major cult centre that was presumably the overarching ‘focal point’ for the majority of the population in a particular region. Hence, advances occurred in sacred architecture closely mirror/parallel developments in early urban town culture, spatial (re)organisation, and social complexity [61,152]. An integrated analysis of the phenomenon of donkey burials remains from archaeological excavations allows for a more comprehensive understanding of decentralised religious practices, ‘history making’ of household descent groups, identities, and symbols of nonelite behaviour. Compared to the traditional fixation on the mostly empty shells of former EB temples and shrines, which has done little to clarify the nature of religious practice for most of the population in the ancient Near East, this holistic approach expands understanding early urban ritual behaviour.

## Figures and Tables

**Figure 1 animals-12-01931-f001:**
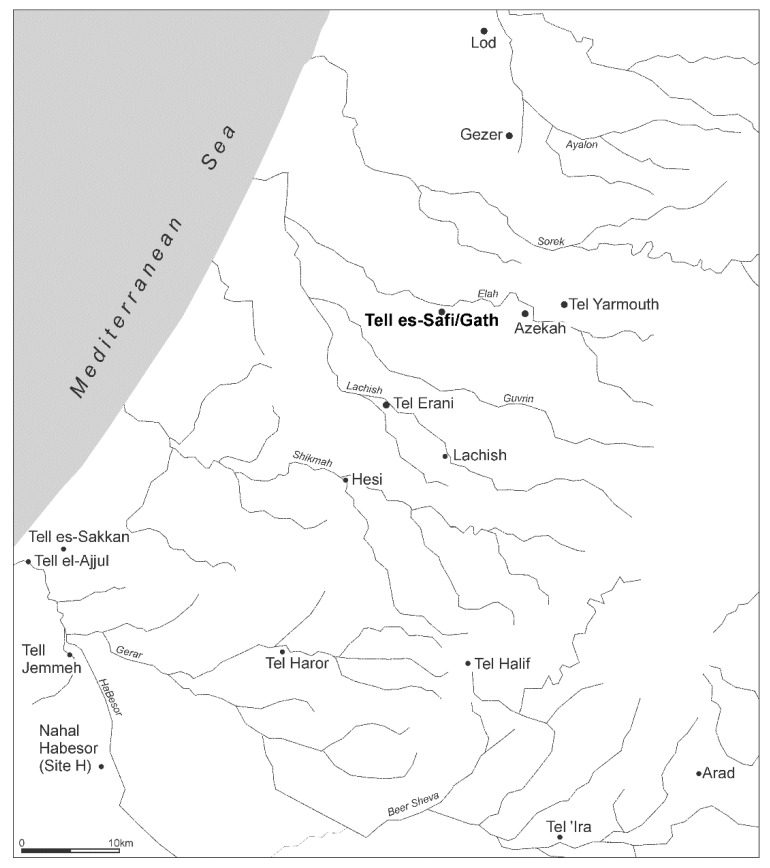
Map showing location of Tell eṣ-Ṣâfi/Gath and some other contemporary important sites in the region. Copyright @ Tell eṣ-Ṣâfi/Gath Archaeological Project.

**Figure 2 animals-12-01931-f002:**
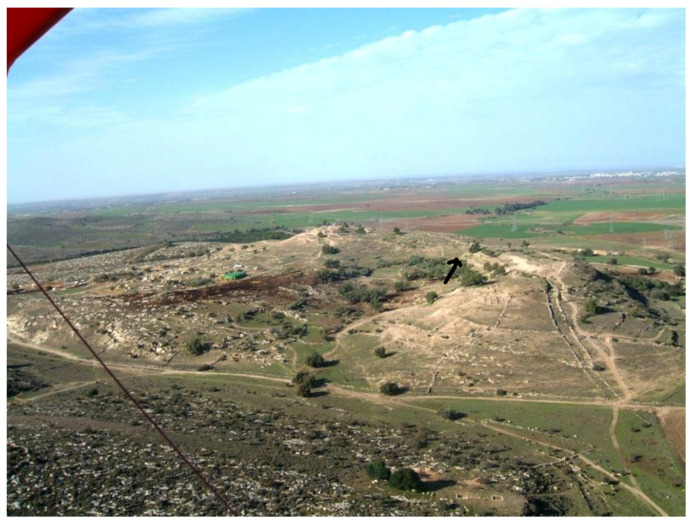
Aerial photograph of Tell eṣ-Ṣâfi/Gath. Area E is located on the east facing slope, as noted by arrow. Copyright @ Tell eṣ-Ṣâfi/Gath Archaeological Project.

**Figure 3 animals-12-01931-f003:**
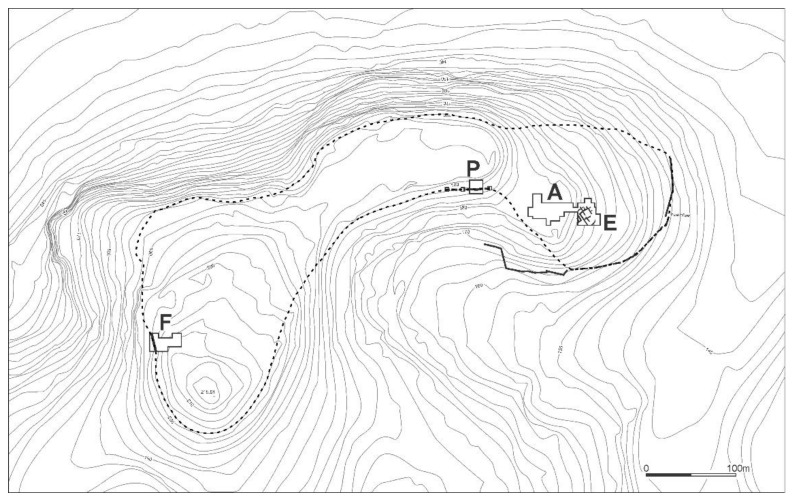
Map showing location of excavation areas and outline of EB fortification and city area. Copyright @ Tell eṣ-Ṣâfi/Gath Archaeological Project.

**Figure 11 animals-12-01931-f011:**
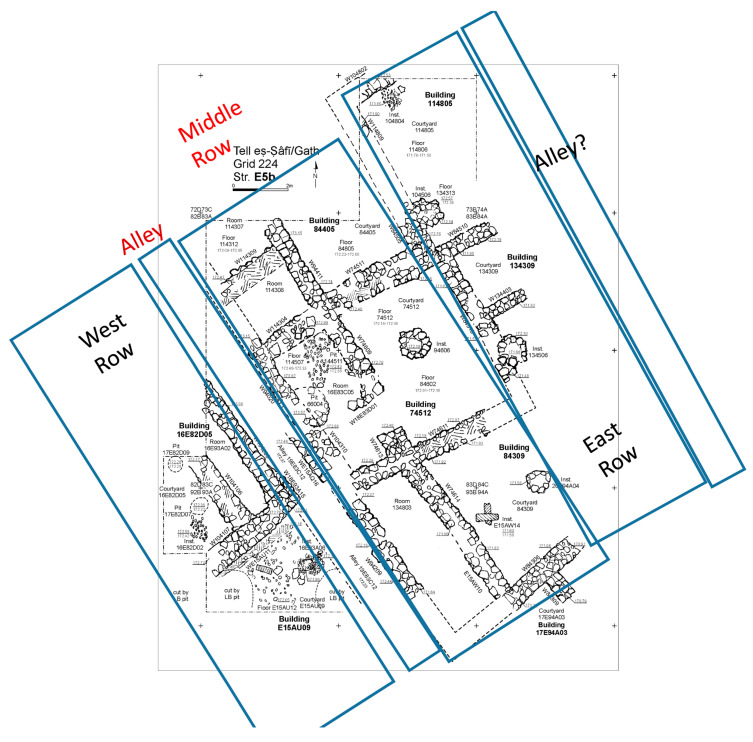
Plan of the EB III neighbourhood uncovered in Stratum E5b of Area E, Tell eṣ-Ṣâfi/Gath, showing the three parallel rows of building, separated by a narrow alleyway. Copyright @ Tell eṣ-Ṣâfi/Gath Archaeological Project.

**Figure 12 animals-12-01931-f012:**
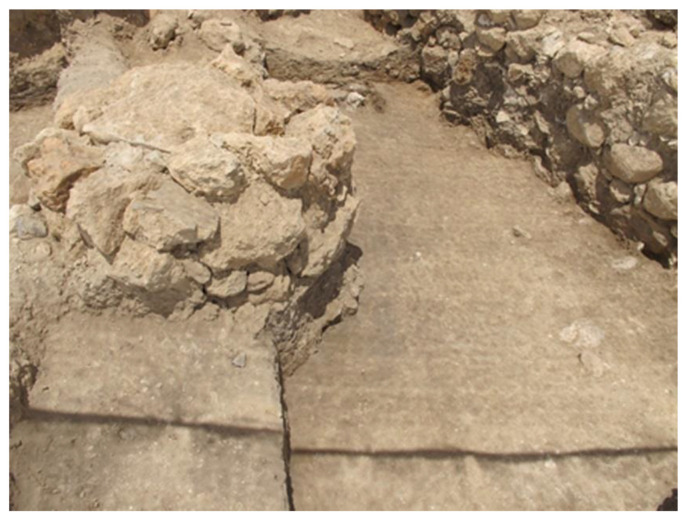
Photograph of round solid stone platform from Strata E5b and c of Area E, Tell eṣ-Ṣâfi/Gath (Installation 94606). Copyright @ Tell eṣ-Ṣâfi/Gath Archaeological Project.

**Figure 13 animals-12-01931-f013:**
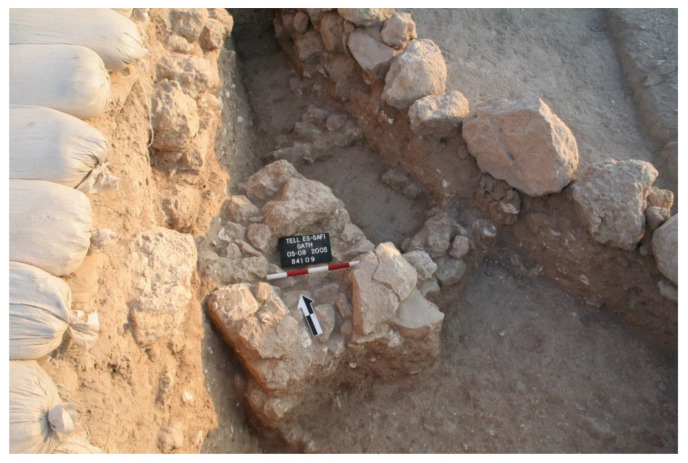
Photograph of solid square stone platform from Strata E5a of Area E, Tell eṣ-Ṣâfi/Gath (Installation 84109). Copyright @ Tell eṣ-Ṣâfi/Gath Archaeological Project.

**Figure 14 animals-12-01931-f014:**
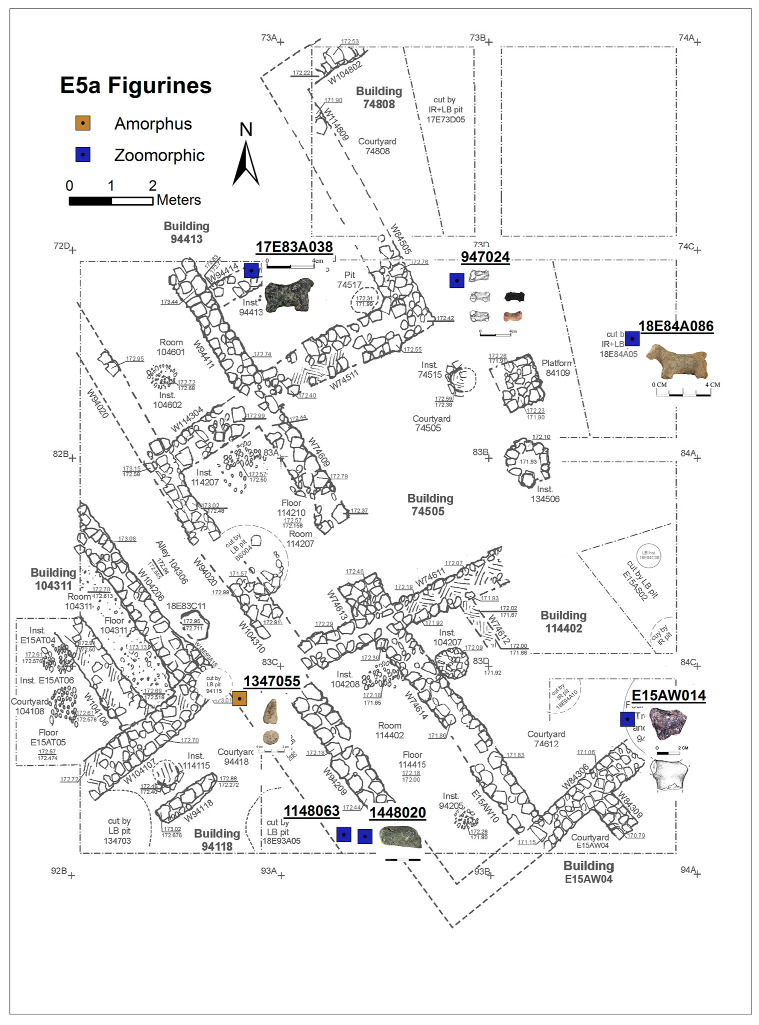
Plan showing the spatial distribution of animal figurines in the EB III neighbourhood uncovered in Stratum E5a of Area E, Tell eṣ-Ṣâfi/Gath. Copyright @ Tell eṣ-Ṣâfi/Gath Archaeological Project.

**Figure 15 animals-12-01931-f015:**
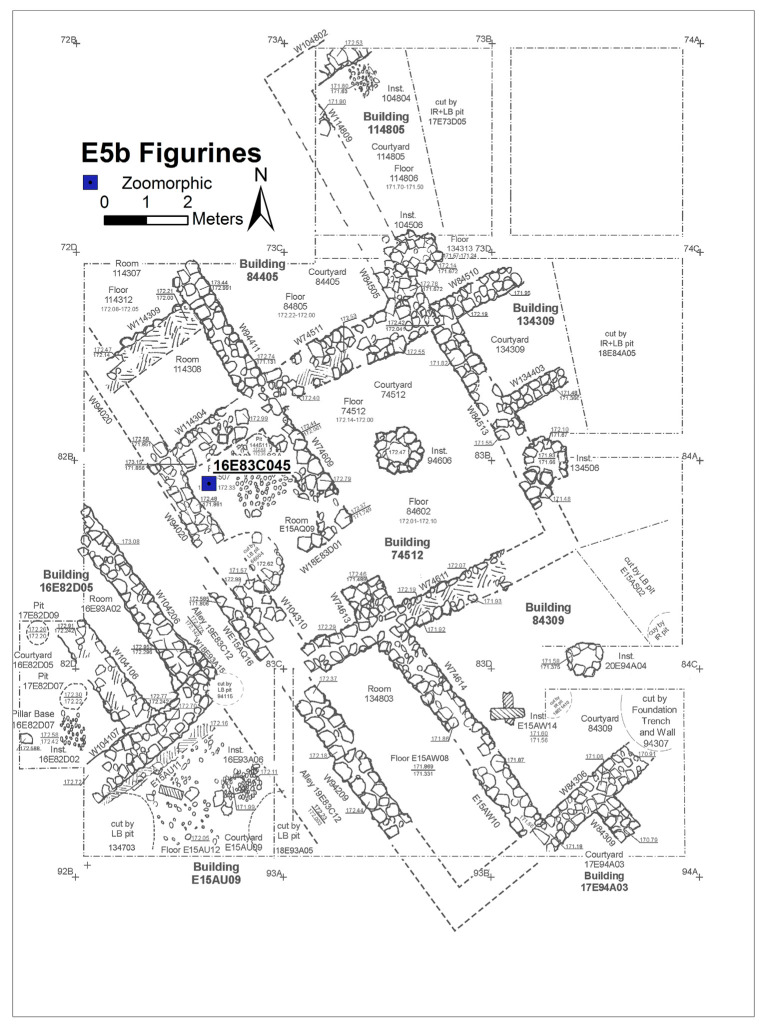
Plan showing the spatial distribution of animal figurines in the EB III neighbourhood uncovered in Stratum E5b of Area E, Tell eṣ-Ṣâfi/Gath. Copyright @ Tell eṣ-Ṣâfi/Gath Archaeological Project.

**Figure 16 animals-12-01931-f016:**
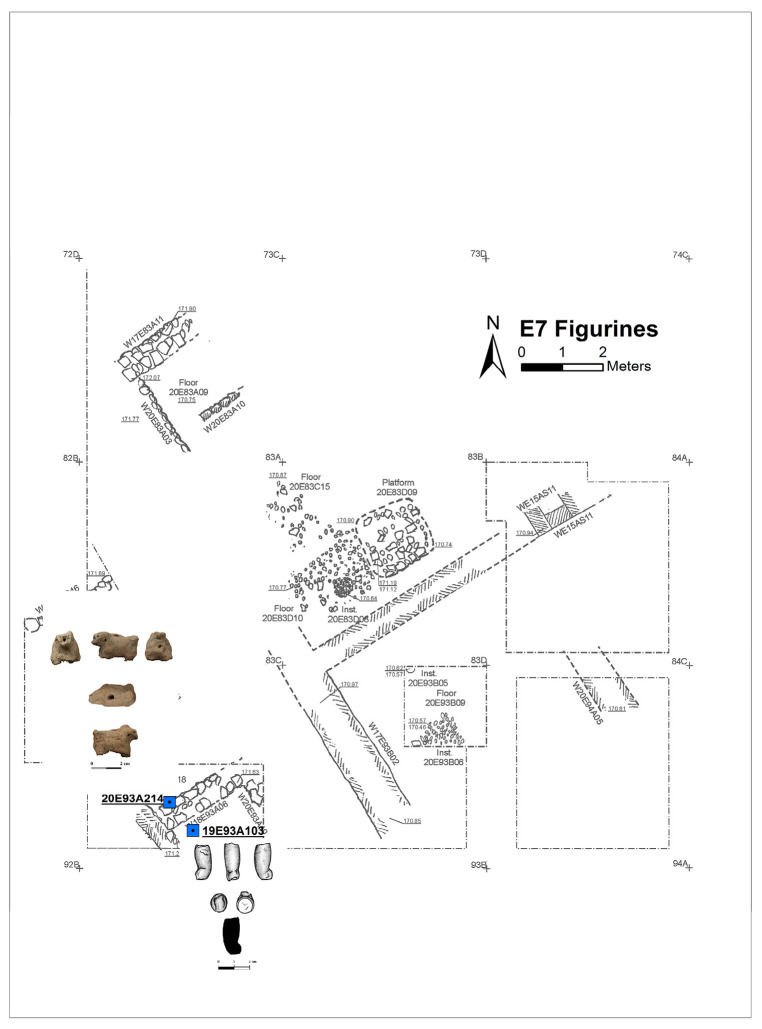
Plan showing the spatial distribution of animal figurines in the EB III neighbourhood uncovered in Stratum E7 of Area E, Tell eṣ-Ṣâfi/Gath. Copyright @ Tell eṣ-Ṣâfi/Gath Archaeological Project.

**Figure 17 animals-12-01931-f017:**
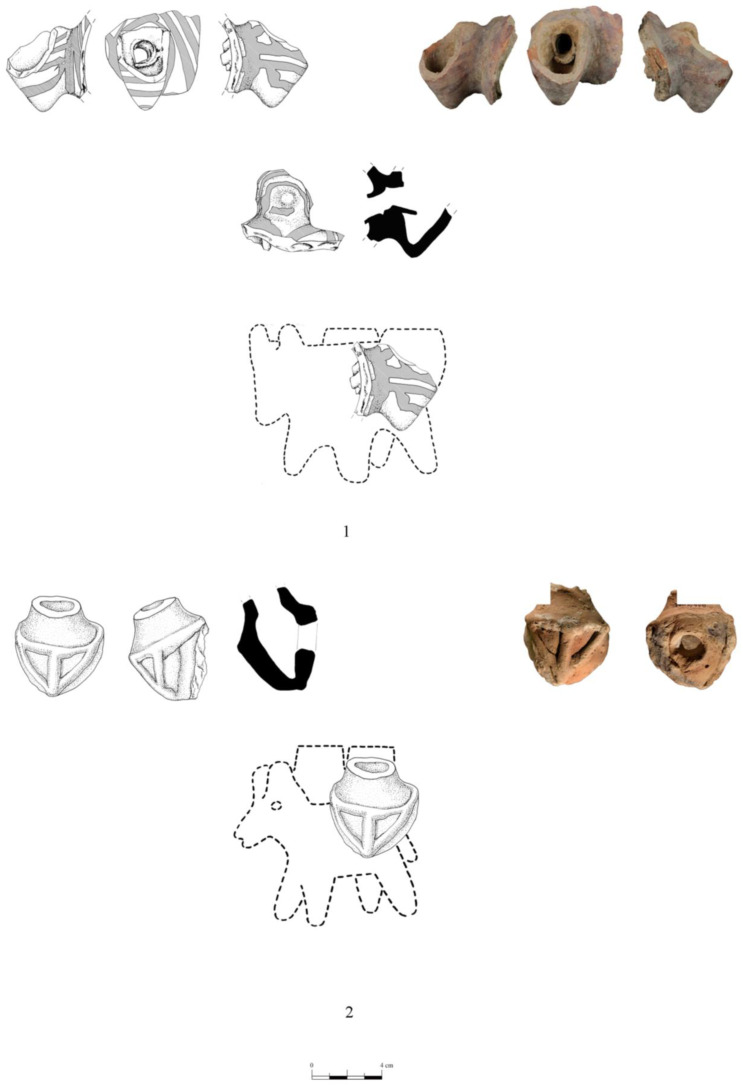
Illustration of fragments of EB donkey figurines found in the EB III neighbourhood of Area E, Tell eṣ-Ṣâfi/Gath. Copyright @ Tell eṣ-Ṣâfi/Gath Archaeological Project.

**Figure 18 animals-12-01931-f018:**
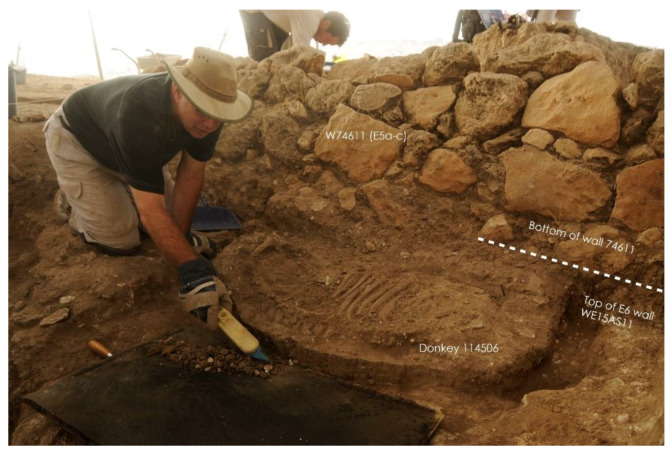
Photograph of the stratigraphic context of Donkey 1 (L114506), showing the location of the shallow pit below the floor of Courtyard 114503 and Wall 74611 of Building 13407 in Stratum E5c of Area E, Tell eṣ-Ṣâfi/Gath. Senior author is shown cleaning the donkey burial. Copyright @ Tell eṣ-Ṣâfi/Gath Archaeological Project and with the permission of Haskel Greenfield.

**Figure 20 animals-12-01931-f020:**
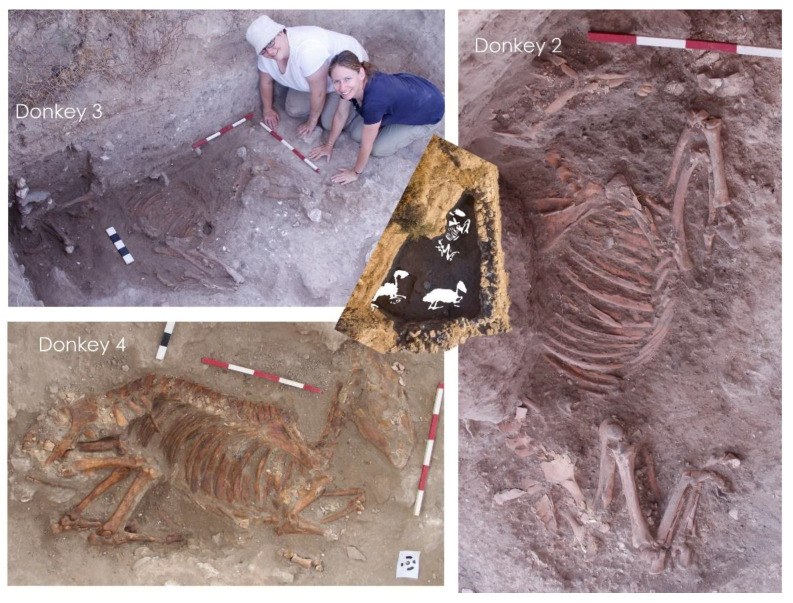
Photographs of the location of the three donkey skeletons (Donkeys 2–4: L19D82D04, L19D83C09, and L20D93A05, respectively) buried below the floor of Courtyard 17E82D02 in Building 17E82D08 in Stratum E5c of Area E, Tell eṣ-Ṣâfi/Gath. Building 17E82D08 was nicknamed ‘the House of Asses’ as a result. Copyright @ Tell eṣ-Ṣâfi/Gath Archaeological Project and with the permission of Elizabeth Arnold and Tina L. Greenfield.

**Figure 21 animals-12-01931-f021:**
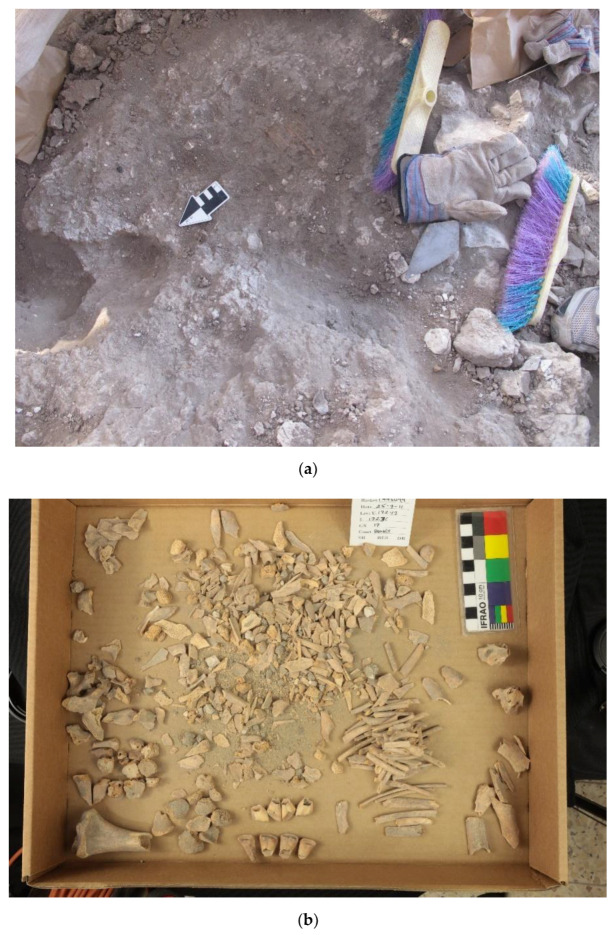
(**a**) in situ photograph of Donkey 5 (L14451) found in a shallow pit below the floor of a complete skeleton of an infant found in a shallow pit below the hearth/cooking installation in Room E15AQ09 in Building 74512 in Stratum E5b of Area E, Tell eṣ-Ṣâfi/Gath, and (**b**) all the fragments after recovery—they fell apart as they were lifted out of the ground, and (**c**) the upper and lower deciduous incisors showing how young the animal was (infant) since they are barely worn. Copyright @ Tell eṣ-Ṣâfi/Gath Archaeological Project.

**Figure 22 animals-12-01931-f022:**
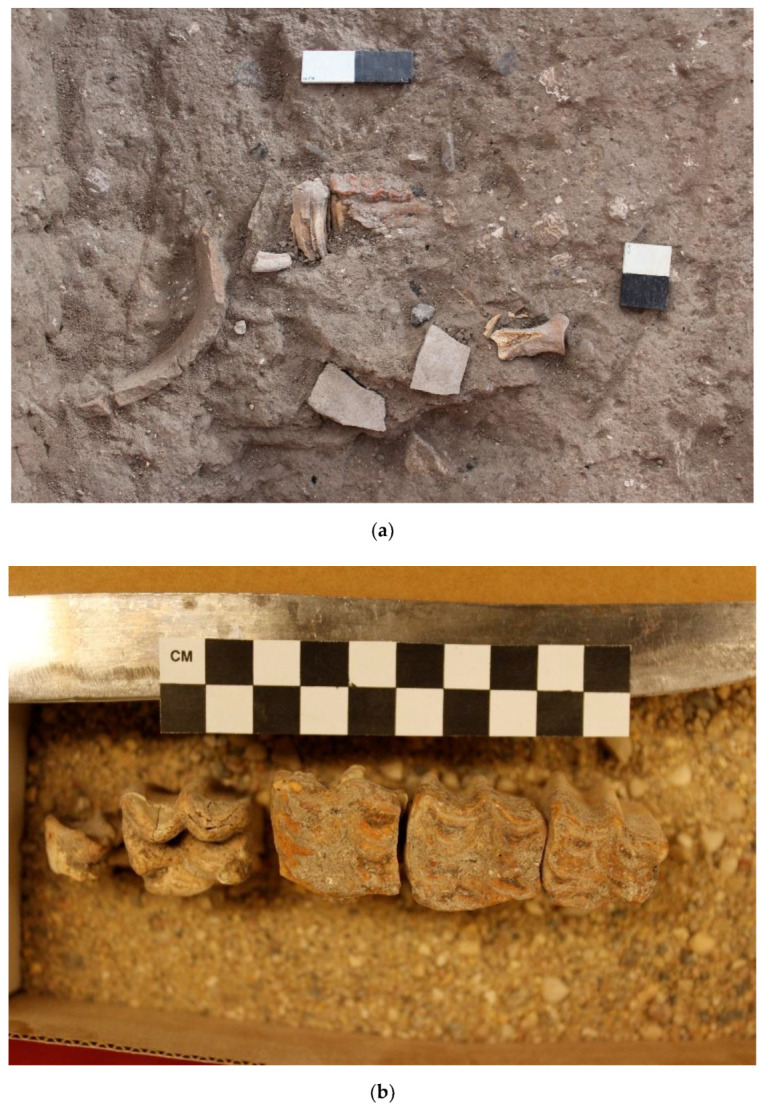
(**a**) Photograph of partial cranium (maxilla and cheek teeth) of Donkey 8 (L17E82D09) being uncovered in a small pit beneath the dirt floor of Courtyard 16E82D05 in Building 1682D05 in Stratum E5b of Area E, Tell eṣ-Ṣâfi/Gath. A few stray bones are also noted. It also points east. (**b**) Photograph of maxillary teeth (upper dm2–4 and M1). Copyright @ Tell eṣ-Ṣâfi/Gath Archaeological Project.

**Figure 23 animals-12-01931-f023:**
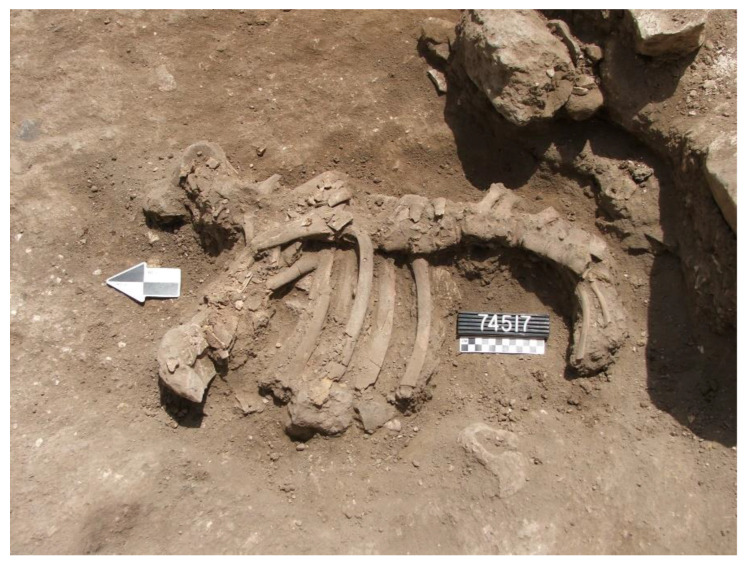
Photograph of Donkey 6 (L74517) buried in a pit beneath the floor of Courtyard 94413 of Building 94413 in Stratum E5a of Area E, Tell eṣ-Ṣâfi/Gath. Arrow is pointing north. Richard Wiskin photo credit for Tell eṣ-Ṣâfi/Gath project. Copyright @ Tell eṣ-Ṣâfi/Gath Archaeological Project.

**Figure 24 animals-12-01931-f024:**
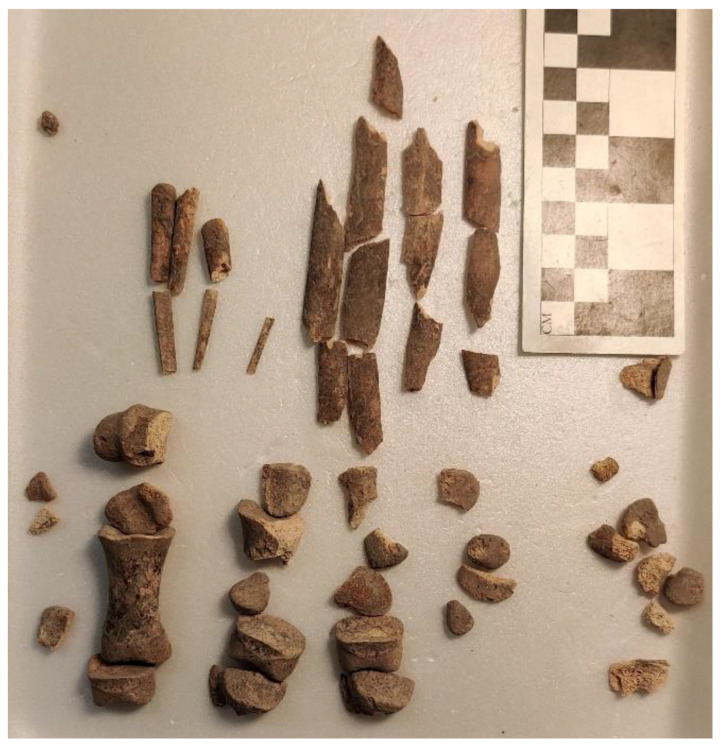
Photograph of the recovered osteological elements from three distal limbs of Donkey 15 (L144506) found extending out of the western balk of Area E. It was buried beneath Wall 104206 of Building 104311 in Stratum E5a, Tell eṣ-Ṣâfi/Gath. Copyright @ Tell eṣ-Ṣâfi/Gath Archaeological Project.

**Figure 25 animals-12-01931-f025:**
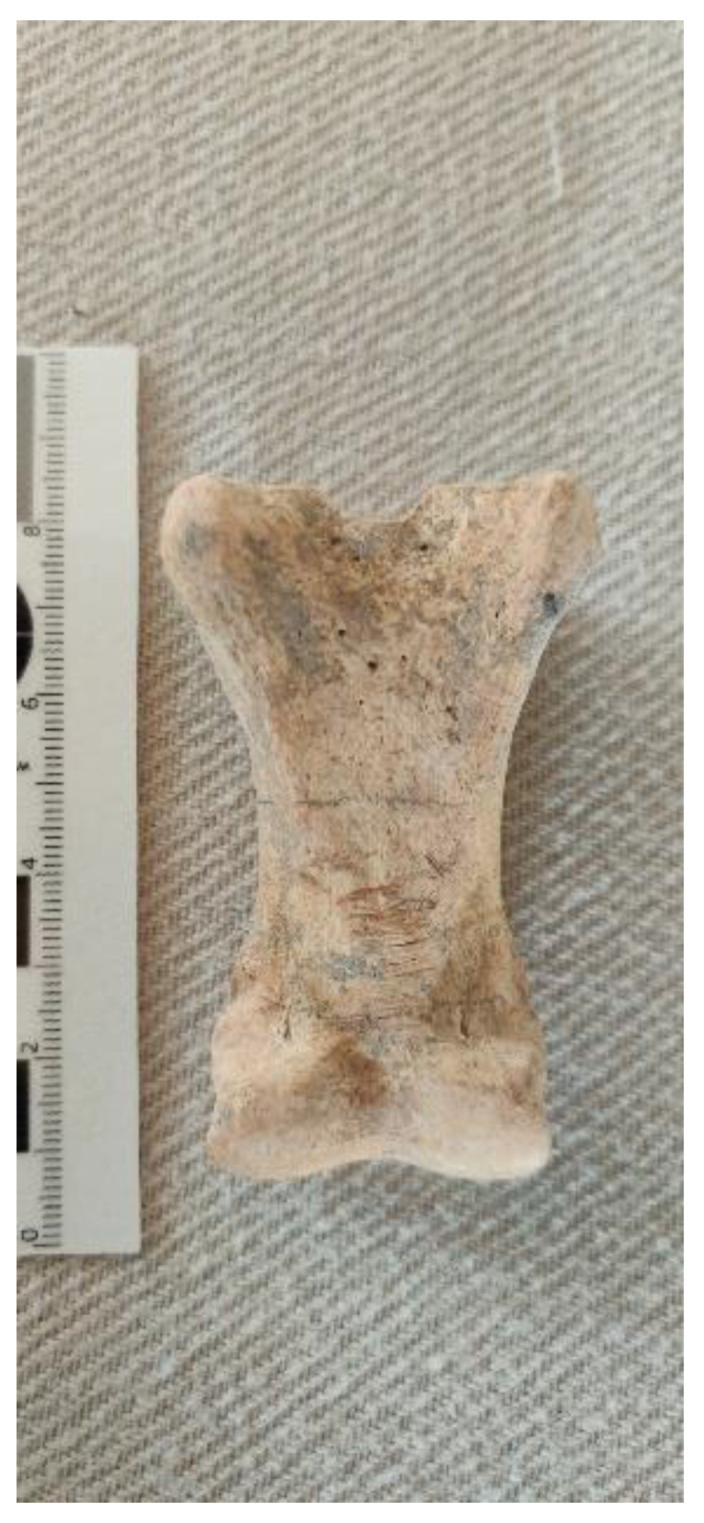
Photograph of slice mark on the first phalange of a donkey (Equus asinus) from L19E83C06 (B19E83C262, Bone 2) from Stratum E5c of Area E, Tell eṣ-Ṣâfi/Gath. Photo credit for Tell eṣ-Ṣâfi/Gath project. Copyright @ Tell eṣ-Ṣâfi/Gath Archaeological Project.

**Figure 26 animals-12-01931-f026:**
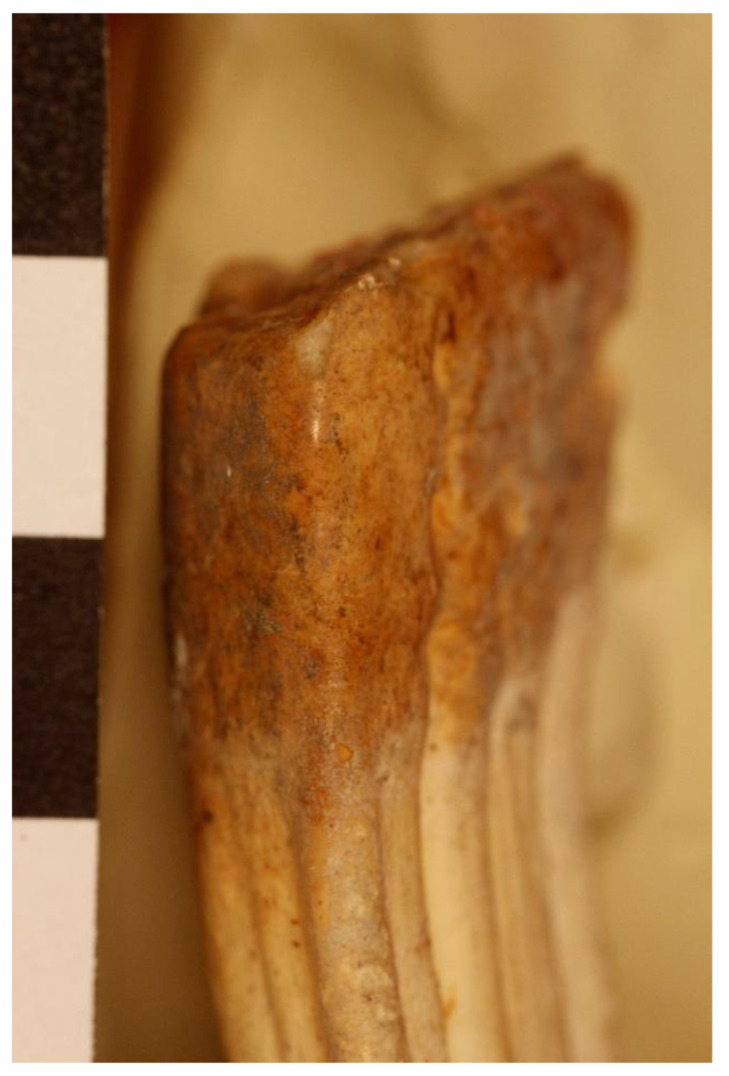
Photograph of Donkey 1’s (L114506) lower right premolar showing bit wear on the mesial face immediately below the occlusal surface. Photo credit for Tell eṣ-Ṣâfi/Gath project. Copyright @ Tell eṣ-Ṣâfi/Gath Archaeological Project.

## Data Availability

Origins faunal remains are curated in the Tell eṣ-Ṣâfi/Gath laboratory at Bar-Ilan University and are available for examination upon request.

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
