# Peer review of "Household Rituals and Merchant Caravanners: The Phenomenon of Early Bronze Age Donkey Burials from Tell eṣ-Ṣâfi/Gath, Israel"

_animals, 2022, doi:10.3390/ani12151931_

Round 1
Reviewer 1 Report
This is an interesting paper, but the donkeys from Tel es-Safi/Gath have been previously discussed in other publications and conference presentations. There is a fair bit of repetition in the text, and I would suggest that the paper could be cut substantially to avoid repetition. I am not sure that all the images are necessary. I would redo Map 1 to mark/emphasize the location of Tel es-Safi/Gath. A shorter and more closely focused paper would be more useful.
I would like a bit more information on how the donkeys themselves were studied. it is clear from the text that there was some mid-identification of the remains in earlier publications. A bit more on how the donkeys were aged and sexed would be useful.
Author Response
- This is an interesting paper, but the donkeys from Tel es-Safi/Gath have been previously discussed in other publications and conference presentations. There is a fair bit of repetition in the text. I would suggest that the paper could be cut substantially to avoid repetition. I am not sure that all the images are necessary.
- HG – I am not sure where the reviewer see repetition. The paper is structured to bring the reader through the data and results. Each image is essential for illustration of the text. This request contradicts the second reviewers glowing comments.
- I would redo Map 1 to mark/emphasize the location of Tel esSafi/Gath.
- HG – We will submit a revised map.
- A shorter and more closely focused paper would be more useful.
- HG – see comment above. This completely contradicts Reviewer 2.
- I would like a bit more information on how the donkeys themselves were studied.
- HG – this is dealt with extensively in the previous publications on the donkeys and would simply add unnecessary additional text.
- it is clear from the text that there was some misidentification of the remains in earlier publications.
- HG – We only indicate that there is a single issue that required correction from an earlier manuscript, and this is already dealt with extensively in Endnote 7.
- A bit more on how the donkeys were aged and sexed would be useful.
- HG - These issues are dealt with extensively in previously publication. It would be redundant to add it to this manuscript, especially since they say there is reported redundancy already. Also, this manuscript is focused on ritual/symbolic behaviour and not on the osteology of the equids.
Reviewer 2 Report
The article is a very well prepared study on the cultural origins of Early Bronze Age donkey skeletal deposits in the Tell eá¹£-á¹¢âfi / Gath stand. Congratulations on the thorough and clear presentation of empirical data, their detailed analysis and interpretation based on multiple premises. An article written in this way is very inspiring.
There is only one comment. Subsection titles such as Architecture as evidence for burials; Installations and platforms as evidence for rituals; Figurines as evidence for rituals and so on are at the same time conclusions which are, after all, the result of the analysis. So appear to pre-judge the evidence being presented in each part. Please, try to rethink titles again.
Numerical citations are used in the journal. Please note that on page 2, line 90 and page 3, line 93, citations are given in (Hodder 2018: 9; Insoll 2005) and (Hawkes 1954), respectively. This needs to be corrected.
Author Response
- The article is a very well prepared study on the cultural origins of Early Bronze Age donkey skeletal deposits in the Tell eá¹£-á¹¢âfi / Gath stand. Congratulations on the thorough and clear presentation of empirical data, their detailed analysis and interpretation based on multiple premises. An article written in this way is very inspiring.
- HG - Thank you :)
- There is only one comment. Subsection titles such as Architecture as evidence for burials; Installations and platforms as evidence for rituals; Figurines as evidence for rituals and so on are at the same time conclusions which are, after all, the result of the analysis. So appear to pre-judge the evidence being presented in each part. Please, try to rethink titles again.
- HG – We have relabeled the subsection titles to reflect the reviewers comments.
- Numerical citations are used in the journal. Please note that on page 2, line 90 and page 3, line 93, citations are given in (Hodder 2018: 9; Insoll 2005) and (Hawkes 1954), respectively. This needs to be corrected.
- HG – Fixed
- We added an additional photograph for Donkey 15 (which was missing in earlier version of the manuscript) and relabeled all subsequent photos.